# SEMI-IMPLICIT VARIATIONAL INFERENCE VIA SCORE MATCHING

**Longlin Yu**
School of Mathematical Sciences
Peking University, Beijing, China
`llyu@pku.edu.cn`

**Cheng Zhang**[*]
School of Mathematical Sciences and Center for Statistical Science
Peking University, Beijing, China
`chengzhang@math.pku.edu.cn`

## ABSTRACT

Semi-implicit variational inference (SIVI) greatly enriches the expressiveness of variational families by considering implicit variational distributions defined in a hierarchical manner. However, due to the intractable densities of variational distributions, current SIVI approaches often use surrogate evidence lower bounds (EL-BOs) or employ expensive inner-loop MCMC runs for direct ELBO maximization for training. In this paper, we propose SIVI-SM, a new method for SIVI based on an alternative training objective via score matching. Leveraging the hierarchical structure of semi-implicit variational families, the score matching objective allows a minimax formulation where the intractable variational densities can be naturally handled with denoising score matching. We show that SIVI-SM closely matches the accuracy of MCMC and outperforms ELBO-based SIVI methods in a variety of Bayesian inference tasks.

## 1 INTRODUCTION

Variational inference(VI) is an approximate Bayesian inference approach where the inference problem is transformed into an optimization problem (Jordan et al., 1999; Wainwright & Jordan, 2008; Blei et al., 2017). It starts by introducing a family of variational distributions over the model parameters (or latent variables) to approximate the posterior. The goal then is to find the closest member from this family of distributions to the target posterior, where the closeness is usually measured by the Kullback-Leibler (KL) divergence from the posterior to the variational approximation. In practice, this is often achieved by maximizing the evidence lower bound (ELBO), which is equivalent to minimizing the KL divergence (Jordan et al., 1999).

One of the classical VI methods is mean-field VI (Bishop & Tipping, 2000), where the variational distributions are assumed to be factorized over the parameters (or latent variables). When combined with conditional conjugacy, this often leads to simple optimization schemes with closed-form update rules (Blei et al., 2017). While popular, the factorizable assumption and conjugacy condition greatly restrict the flexibility and applicability of variational posteriors, especially for complicated models with high dimensional parameter space. Recent years have witnessed much progress in the field of VI that extends it to more complicated settings. For example, the conjugacy condition has been removed by the black-box VI methods which allow a broad class of models via Monte carlo gradient estimators (Nott et al., 2012; Paisley et al., 2012; Ranganath et al., 2014; Rezende et al., 2014; Kingma & Welling, 2014). On the other hand, more flexible variational families have been proposed that either explicitly incorporate more complicated structures among the parameters (Jaakkola & Jordan, 1998; Saul & Jordan, 1996; Giordano et al., 2015; Tran et al., 2015) or borrow ideas from invertible transformation of probability distributions (Rezende & Mohamed, 2015; Dinh et al., 2017; Kingma et al., 2016; Papamakarios et al., 2019). All these methods require tractable densities for the variational distributions.

It turns out that the variational family can be further expanded by allowing implicit models that have intractable densities but are easy to sample from (Huszár, 2017). One way to construct these implicit models is to transform a simple base distribution via a deterministic map, i.e., a deep neural

---

[*]Corresponding author.

network (Tran et al., 2017; Mescheder et al., 2017; Shi et al., 2018a;b; Song et al., 2019). Due to the intractable densities of implicit models, when evaluating the ELBO during training, one often resorts to density ratio estimation which is known to be challenging in high-dimensional settings (Sugiyama et al., 2012). To avoid density ratio estimation, semi-implicit variational inference (SIVI) has been proposed where the variational distributions are formed through a semi-implicit hierarchical construction and surrogate ELBOs (asymptotically unbiased) are employed for training (Yin & Zhou, 2018; Moens et al., 2021). Instead of surrogate ELBOs, an unbiased gradient estimator of the exact ELBO has been derived based on MCMC samples from a reverse conditional (Titsias & Ruiz, 2019). However, the computation for the inner-loop MCMC runs can easily become expensive in high-dimensional regimes. There are also approaches that estimate the gradients instead of the objective (Li & Turner, 2018; Shi et al., 2018b; Song et al., 2019).

Besides KL divergence, score-based distance measures have also been introduced in various statistical tasks (Hyvärinen, 2005; Zhang et al., 2018) and have shown advantages in complicated nonlinear models (Song & Ermon, 2019; Ding et al., 2019; Elkhalil et al., 2021). Recently, there are also some studies that use score matching for variational inference (Yang et al., 2019; Hu et al., 2018). However, these methods are not designed for SIVI and hence either do not apply to or can not fully exploit the hierarchical structure of semi-implicit variational distributions. In this paper, we propose SIVI-SM, a new method for SIVI using an alternative training objective via score matching. We show that the score matching objective and the semi-implicit hierarchical construction of variational posteriors can be combined in a minimax formulation where the intractability of densities is naturally handled with denoising score matching. We demonstrate the effectiveness and efficiency of our method on both synthetic distributions and a variety of real data Bayesian inference tasks.

## 2 BACKGROUND

**Semi-Implicit Variational Inference**    Semi-implicit variational inference (SIVI) (Yin & Zhou, 2018) posits a flexible variational family defined hierarchically using a mixing parameter as follows

$$\boldsymbol{x} \sim q_\phi(\boldsymbol{x}|\boldsymbol{z}), \quad \boldsymbol{z} \sim q_\xi(\boldsymbol{z}), \quad q_\varphi(\boldsymbol{x}) = \int q_\phi(\boldsymbol{x}|\boldsymbol{z}) q_\xi(\boldsymbol{z}) d\boldsymbol{z}. \tag{1}$$

where $\varphi = \{\phi, \xi\}$ are the variational parameters. This variational distribution is called semi-implicit as the conditional layer $q_\phi(\boldsymbol{x}|\boldsymbol{z})$ is required to be explicit but the mixing layer $q_\xi(\boldsymbol{z})$ can be implicit, and $q_\varphi(\boldsymbol{x})$ is often implicit unless $q_\xi(\boldsymbol{z})$ is conjugate to $q_\phi(\boldsymbol{x}|\boldsymbol{z})$. Compared to standard VI, the above hierarchical construction allows a much richer variational family that is able to capture complicated dependencies between parameters (Yin & Zhou, 2018).

Similar to standard VI, current SIVI methods fit the model parameters by maximizing the evidence lower bound (ELBO) derived as follows

$$\log p(D) \geq \text{ELBO} := \mathbb{E}_{\boldsymbol{x} \sim q_\varphi(\boldsymbol{x})} \left[ \log p(D, \boldsymbol{x}) - \log q_\varphi(\boldsymbol{x}) \right],$$

where $D$ is the observed data. As $q_\varphi(\boldsymbol{x})$ is no longer tractable, Yin & Zhou (2018) considered a sequence of lower bounds of ELBO

$$\text{ELBO} \geq \mathcal{L}_L := \mathbb{E}_{\boldsymbol{z} \sim q_\xi(\boldsymbol{z}), \boldsymbol{x} \sim q_\phi(\boldsymbol{x}|\boldsymbol{z})} \mathbb{E}_{\boldsymbol{z}^{(1)}, \cdots, \boldsymbol{z}^{(L)} \overset{i.i.d.}{\sim} q_\xi(\boldsymbol{z})} \log \frac{p(D, \boldsymbol{x})}{\frac{1}{L+1} \left( q_\phi(\boldsymbol{x}|\boldsymbol{z}) + \sum_{l=1}^L q_\phi(\boldsymbol{x}|\boldsymbol{z}^{(l)}) \right)}.$$

Note that $\mathcal{L}_L$ is an asymptotically exact surrogate ELBO as $L \to \infty$. An increasing sequence of $\{L_t\}_{t=1}^\infty$, therefore, is often suggested, with $\mathcal{L}_{L_t}$ being optimized at the $t$-$th$ iteration.

Instead of maximizing surrogate ELBOs, Titsias & Ruiz (2019) proposed unbiased implicit variational inference (UIVI) which is based on an unbiased gradient estimator of the exact ELBO. More specifically, consider a fixed mixing distribution $q_\xi(\boldsymbol{z}) = q(\boldsymbol{z})$ and a reparameterizable conditional $q_\phi(\boldsymbol{x}|\boldsymbol{z})$ such that $\boldsymbol{x} = T_\phi(\boldsymbol{z}, \boldsymbol{\epsilon}), \boldsymbol{\epsilon} \sim q_\epsilon(\boldsymbol{\epsilon}) \Leftrightarrow \boldsymbol{x} \sim q_\phi(\boldsymbol{x}|\boldsymbol{z})$, then

$$\nabla_\phi \text{ELBO} = \nabla_\phi \mathbb{E}_{q_\epsilon(\boldsymbol{\epsilon}) q(\boldsymbol{z})} \left[ \log p(D, \boldsymbol{x}) - \log q_\phi(\boldsymbol{x})|_{\boldsymbol{x}=T_\phi(\boldsymbol{z}, \boldsymbol{\epsilon})} \right],$$

$$:= \mathbb{E}_{q_\epsilon(\boldsymbol{\epsilon}) q(\boldsymbol{z})} \left[ g_\phi^{mod}(\boldsymbol{z}, \boldsymbol{\epsilon}) + g_\phi^{ent}(\boldsymbol{z}, \boldsymbol{\epsilon}) \right], \tag{2}$$

where

$$g_\phi^{mod}(\boldsymbol{z}, \boldsymbol{\epsilon}) := \nabla_{\boldsymbol{x}} \log p(D, \boldsymbol{x})|_{\boldsymbol{x}=T_\phi(\boldsymbol{z}, \boldsymbol{\epsilon})} \nabla_\phi T_\phi(\boldsymbol{z}, \boldsymbol{\epsilon}), \tag{3}$$

$$g_\phi^{ent}(\boldsymbol{z}, \boldsymbol{\epsilon}) := - \mathbb{E}_{q_\phi(\boldsymbol{z}'|\boldsymbol{x})} \nabla_{\boldsymbol{x}} \log q_\phi(\boldsymbol{x}|\boldsymbol{z}')|_{\boldsymbol{x}=T_\phi(\boldsymbol{z}, \boldsymbol{\epsilon})} \nabla_\phi T_\phi(\boldsymbol{z}, \boldsymbol{\epsilon}). \tag{4}$$

The gradient term in 4 involves an expectation w.r.t. the reverse conditional $q_\phi(\boldsymbol{z}|\boldsymbol{x})$ which can be estimated using an MCMC sampler (e.g., Hamiltonian Monte Carlo (Neal, 2011)). However, the inner-loop MCMC runs can easily become computationally expensive in high dimensional regimes. See a more detailed discussion on the derivation and computation issues of UIVI in Appendix A and D.

**Score Matching**  Score matching is first introduced by Hyvärinen (2005) to learn un-normalized statistical models given i.i.d. samples from an unknown data distribution $p(\boldsymbol{x})$. Instead of estimating $p(\boldsymbol{x})$ directly, score matching trains a score network $S(\boldsymbol{x})$ to estimate the score of the data distribution, i.e. $\nabla \log p(\boldsymbol{x})$, by minimizing the score matching objective $\mathbb{E}_{p(\boldsymbol{x})}[\frac{1}{2}\|S(\boldsymbol{x}) - \nabla_{\boldsymbol{x}} \log p(\boldsymbol{x})\|_2^2]$. Using the trick of partial integration, Hyvärinen (2005) shows that the score matching objective is equivalent to the following up to a constant

$$\mathbb{E}_{\boldsymbol{x} \sim p(\boldsymbol{x})}[\mathrm{Tr}(\nabla_{\boldsymbol{x}}(S(\boldsymbol{x}))) + \frac{1}{2}\|S(\boldsymbol{x})\|_2^2]. \tag{5}$$

The expectation in Eq. 5 can be quickly estimated using data samples. However, it is often challenging to scale up score matching to high dimensional data due to the computation of $\mathrm{Tr}\,\nabla_{\boldsymbol{x}}(S(\boldsymbol{x}))$.

A commonly used variant of score matching that can scale up to high dimensional data is denoising score matching (DSM) (Vincent, 2011). The first step of DSM is to perturb the data with a known noise distribution $q_\sigma(\tilde{\boldsymbol{x}}|\boldsymbol{x})$, which leads to a perturbed data distribution $q_\sigma(\tilde{\boldsymbol{x}}) = \int q_\sigma(\tilde{\boldsymbol{x}}|\boldsymbol{x})p(\boldsymbol{x})d\boldsymbol{x}$. The score matching objective for $q_\sigma(\tilde{\boldsymbol{x}})$ turns out to be equivalent to the following up to a constant

$$\frac{1}{2}\mathbb{E}_{q_\sigma(\tilde{\boldsymbol{x}}|\boldsymbol{x})p(\boldsymbol{x})} \left[\|S(\tilde{\boldsymbol{x}}) - \nabla_{\tilde{\boldsymbol{x}}} \log q_\sigma(\tilde{\boldsymbol{x}}|\boldsymbol{x})\|_2^2\right]. \tag{6}$$

Unlike Eq. 5, Eq. 6 does not involve the trace term and can be computed efficiently, as long as the score of the noise distribution $\nabla_{\tilde{\boldsymbol{x}}} \log q_\sigma(\tilde{\boldsymbol{x}}|\boldsymbol{x})$ is easy to compute. Note that the optimal score network here estimates the score of the perturbed data distribution rather than that of the true data distribution. A small noise, therefore, is required for accurate approximation of the true data score $\nabla \log p(\boldsymbol{x})$. Despite this, DSM is widely used in learning energy based models (Saremi et al., 2018) and score based generative models (Song & Ermon, 2019).

## 3 PROPOSED METHOD

While ELBO-based training objectives prove effective for semi-implicit variational inference, current approaches either rely on surrogates of the exact ELBO or expensive inner-loop MCMC runs for unbiased gradient estimates due to the intractable variational posteriors. In this section, we introduce an alternative training objective for SIVI based on score matching. We show that the score matching objective can be reformulated in a minimax fashion such that the semi-implicit hierarchical construction of variational posteriors can be efficiently exploited using denoising score matching. Throughout this section, we assume the conditional layer $q_\phi(\boldsymbol{x}|\boldsymbol{z})$ to be reparameterizable and its score function $\nabla_{\boldsymbol{x}} \log q_\phi(\boldsymbol{x}|\boldsymbol{z})$ is easy to evaluate[1].

### 3.1 A MINIMAX REFORMULATION

Rather than maximizing the ELBO as in previous semi-implicit variational inference methods, we can instead minimize the following Fisher divergence that compares the score functions of the target and the variational distribution

$$\min_\varphi \quad \mathbb{E}_{\boldsymbol{x} \sim q_\varphi(\boldsymbol{x})}\|S(\boldsymbol{x}) - \nabla_{\boldsymbol{x}} \log q_\varphi(\boldsymbol{x})\|_2^2. \tag{7}$$

---

[1]This assumption is quite general and it holds for many classical distributions that are commonly used as conditionals, such as Gaussian and other exponential family distributions.

Here $S(\boldsymbol{x}) = \nabla_{\boldsymbol{x}} \log p(\boldsymbol{x}|D) = \nabla_{\boldsymbol{x}} \log p(D, \boldsymbol{x})$ is the score of the target posterior distribution, and the variational distribution $q_\varphi(\boldsymbol{x})$ is defined in Eq. 1. Due to the semi-implicit construction in Eq. 1, the score of variational distribution, i.e. $\nabla_{\boldsymbol{x}} \log q_\varphi(\boldsymbol{x})$, is intractable, making the Fisher divergence in Eq. 7 not readily computable. Although the hierarchical structure of $q_\varphi(\boldsymbol{x})$ allows us to estimate its score function via denoising score matching, the estimated score function would break the dependency on the variational parameter $\varphi$, leading to biased gradient estimates (see an illustration in Appendix L). Fortunately, this issue can be remedied by reformulating 7 as a minimax problem. The key observation is that the squared norm of $S(\boldsymbol{x}) - \nabla_{\boldsymbol{x}} \log q_\varphi(\boldsymbol{x})$ can be viewed as the maximum value of the following nested optimization problem

$$\|S(\boldsymbol{x}) - \nabla_{\boldsymbol{x}} \log q_\varphi(\boldsymbol{x})\|_2^2 = \max_{f(\boldsymbol{x})} \quad 2f(\boldsymbol{x})^T[S(\boldsymbol{x}) - \nabla_{\boldsymbol{x}} \log q_\varphi(\boldsymbol{x})] - \|f(\boldsymbol{x})\|_2^2, \quad \forall \boldsymbol{x}.$$

where $f(\boldsymbol{x})$ is an arbitrary function of $\boldsymbol{x}$, and the unique optimal solution is

$$f^\varphi(\boldsymbol{x}) := S(\boldsymbol{x}) - \nabla_{\boldsymbol{x}} \log q_\varphi(\boldsymbol{x}).$$

Based on this observation, we can rewrite the optimization problem in 7 as

$$\min_\varphi \max_f \quad \mathbb{E}_{\boldsymbol{x} \sim q_\varphi(\boldsymbol{x})} 2f(\boldsymbol{x})^T[S(\boldsymbol{x}) - \nabla_{\boldsymbol{x}} \log q_\varphi(\boldsymbol{x})] - \|f(\boldsymbol{x})\|_2^2. \tag{8}$$

Now we can take advantage of the hierarchical structure of $q_\varphi(\boldsymbol{x})$ to get ride of the intractable score term $\nabla_{\boldsymbol{x}} \log q_\varphi(\boldsymbol{x})$ in Eq. 8, similarly as done in DSM. More specifically, note that

$$\nabla_{\boldsymbol{x}} \log q_\varphi(\boldsymbol{x}) = \frac{1}{q_\varphi(\boldsymbol{x})} \int q_\xi(\boldsymbol{z}) q_\phi(\boldsymbol{x}|\boldsymbol{z}) \nabla_{\boldsymbol{x}} \log q_\phi(\boldsymbol{x}|\boldsymbol{z}) d\boldsymbol{z}.$$

We have

$$\mathbb{E}_{\boldsymbol{x} \sim q_\varphi(\boldsymbol{x})} f(\boldsymbol{x})^T \nabla_{\boldsymbol{x}} \log q_\varphi(\boldsymbol{x}) = \int q_\varphi(\boldsymbol{x}) f(\boldsymbol{x})^T \frac{1}{q_\varphi(\boldsymbol{x})} \int q_\xi(\boldsymbol{z}) q_\phi(\boldsymbol{x}|\boldsymbol{z}) \nabla_{\boldsymbol{x}} \log q_\phi(\boldsymbol{x}|\boldsymbol{z}) d\boldsymbol{z} d\boldsymbol{x},$$

$$= \iint q_\xi(\boldsymbol{z}) q_\phi(\boldsymbol{x}|\boldsymbol{z}) f(\boldsymbol{x})^T \nabla_{\boldsymbol{x}} \log q_\phi(\boldsymbol{x}|\boldsymbol{z}) d\boldsymbol{z} d\boldsymbol{x},$$

$$= \mathbb{E}_{\boldsymbol{z} \sim q_\xi(\boldsymbol{z}), \boldsymbol{x} \sim q_\phi(\boldsymbol{x}|\boldsymbol{z})} f(\boldsymbol{x})^T \nabla_{\boldsymbol{x}} \log q_\phi(\boldsymbol{x}|\boldsymbol{z}). \tag{9}$$

Substituting Eq. 9 into Eq. 8 completes our reformulation.

**Theorem 1.** *Assume the variational distribution $q_\varphi(\boldsymbol{x})$ is a semi-implicit distribution defined by Eq. 1, then the optimization problem in 7 is equivalent to the following minimax problem*

$$\min_\varphi \max_f \quad \mathbb{E}_{\boldsymbol{z} \sim q_\xi(\boldsymbol{z}), \boldsymbol{x} \sim q_\phi(\boldsymbol{x}|\boldsymbol{z})} 2f(\boldsymbol{x})^T[S(\boldsymbol{x}) - \nabla_{\boldsymbol{x}} \log q_\phi(\boldsymbol{x}|\boldsymbol{z})] - \|f(\boldsymbol{x})\|_2^2, \tag{10}$$

*Where $\varphi = \{\phi, \xi\}$. Moreover, assume that $f$ can represent any function. If $(\varphi^*, f^*)$ defines a Nash-equilibrium of Eq. 10, then $f^*, \varphi^*$ is given by*

$$f^*(\boldsymbol{x}) = S(\boldsymbol{x}) - \nabla_{\boldsymbol{x}} \log q_{\varphi^*}(\boldsymbol{x}),$$

$$\varphi^* \in \arg\min_\varphi \{\mathbb{E}_{\boldsymbol{x} \sim q_\varphi(\boldsymbol{x})} \|S(\boldsymbol{x}) - \nabla_{\boldsymbol{x}} \log q_\varphi(\boldsymbol{x})\|_2^2\}. \tag{11}$$

See a detailed proof of Theorem 1 in Appendix B. Note that the objective in equation 8 can also be derived via Stein discrepancy minimization (Liu et al., 2016; Gorham & Mackey, 2015; Ranganath et al., 2016; Grathwohl et al., 2020). However, our reformulation in equation 10 takes a further step by utilizing the hierarchical structure of $q_\varphi(\boldsymbol{x})$ and hence can easily scale up to high dimensions. See Appendix K for a more detailed discussion.

## 3.2 PRACTICAL ALGORITHMS

In practice, we parameterize $f$ with a neural network $f_\psi(\boldsymbol{x})$. According to the above minimax reformulation, the Monte Carlo estimation of the objective function in Eq. 10 can be easily obtained by sampling from the hierarchical variational distribution $\boldsymbol{z} \sim q_\xi(\boldsymbol{z}), \boldsymbol{x} \sim q_\phi(\boldsymbol{x}|\boldsymbol{z})$. Furthermore, using the reparameterization trick (Kingma & Welling, 2013), i.e. $\boldsymbol{x} = T_\phi(\boldsymbol{z}; \boldsymbol{\epsilon}), \boldsymbol{z} = h_\xi(\boldsymbol{\gamma})$, where $\boldsymbol{\epsilon} \sim q_\epsilon(\boldsymbol{\epsilon}), \boldsymbol{\gamma} \sim q_\gamma(\boldsymbol{\gamma})$, we can rewrite Eq. 10 as follows

$$\min_\varphi \max_\psi \quad \mathbb{E}_{q_\gamma(\boldsymbol{\gamma}), q_\epsilon(\boldsymbol{\epsilon})} \left[ 2f_\psi(\boldsymbol{x})^T[S(\boldsymbol{x}) - \nabla_{\boldsymbol{x}} \log q_\phi(\boldsymbol{x}|\boldsymbol{z}) - \frac{1}{2} f_\psi(\boldsymbol{x})] \Big|_{\boldsymbol{x} = T_\phi(\boldsymbol{z}, \boldsymbol{\epsilon}), \boldsymbol{z} = h_\xi(\boldsymbol{\gamma})} \right],$$

---

**Algorithm 1** SIVI-SM with multivariate Gaussian conditional layer

---

**Input:** Score of target posterior distribution $S(\boldsymbol{x})$. Total iteration number $N$. Number of gradient steps $K$ for the inner optimization on $f_\psi(\boldsymbol{x})$.
**Output:** Variational parameter $\varphi$ and the neural network parameter $\psi$.
Initialize $\varphi_0, \psi_0$
**for** $t = 0$ **to** $N-1$ **do**
    Sample $\{\boldsymbol{\gamma}^{(1)}, \boldsymbol{\gamma}^{(2)}, \cdots, \boldsymbol{\gamma}^{(m)}\}$ from prior $q_{\boldsymbol{\gamma}}(\boldsymbol{\gamma})$ and let $\boldsymbol{z}^{(i)} = h_\xi(\boldsymbol{\gamma}^{(i)}), i = 1, \ldots, m$.
    Sample $\{\boldsymbol{\epsilon}^{(1)}, \boldsymbol{\epsilon}^{(2)}, \cdots, \boldsymbol{\epsilon}^{(m)}\}$ from $\mathcal{N}(0, \boldsymbol{I})$.
    Compute $\boldsymbol{x}^{(i)} = \boldsymbol{\mu}_\phi(\boldsymbol{z}^{(i)}) + \boldsymbol{\sigma}_\phi(\boldsymbol{z}^{(i)}) \odot \boldsymbol{\epsilon}^{(i)}$.
    Update $\varphi$ by descending its stochastic gradient:

$$\nabla_\varphi \frac{1}{m} \sum_{i=1}^m f_\psi(\boldsymbol{x}^{(i)})^T [S(\boldsymbol{x}^{(i)}) + \boldsymbol{\sigma}_\phi(\boldsymbol{z}^{(i)})^{-1} \odot \boldsymbol{\epsilon}^{(i)}] - \frac{1}{2} \|f_\psi(\boldsymbol{x}^{(i)})\|_2^2.$$

**for** $j = 1$ **to** $K$ **do**
    Sample $\{\boldsymbol{\gamma}^{(1)}, \boldsymbol{\gamma}^{(2)}, \cdots, \boldsymbol{\gamma}^{(m)}\}$ from prior $q_{\boldsymbol{\gamma}}(\boldsymbol{\gamma})$ and let $\boldsymbol{z}^{(i)} = h_\xi(\boldsymbol{\gamma}^{(i)}), i = 1, \ldots, m$.
    Sample $\{\boldsymbol{\epsilon}^{(1)}, \boldsymbol{\epsilon}^{(2)}, \cdots, \boldsymbol{\epsilon}^{(m)}\}$ from $\mathcal{N}(0, \boldsymbol{I})$.
    Compute $\boldsymbol{x}^{(i)} = \boldsymbol{\mu}_\phi(\boldsymbol{z}^{(i)}) + \boldsymbol{\sigma}_\phi(\boldsymbol{z}^{(i)}) \odot \boldsymbol{\epsilon}^{(i)}$.
    Update $\psi$ by ascending its stochastic gradient:

$$\nabla_\psi \frac{1}{m} \sum_{i=1}^m f_\psi(\boldsymbol{x}^{(i)})^T [S(\boldsymbol{x}^{(i)}) + \boldsymbol{\sigma}_\phi(\boldsymbol{z}^{(i)})^{-1} \odot \boldsymbol{\epsilon}^{(i)}] - \frac{1}{2} \|f_\psi(\boldsymbol{x}^{(i)})\|_2^2.$$

**end for**
**end for**

---

This allows us to directly optimize the parameters $\varphi, \psi$ with gradient based optimization methods (Goodfellow et al., 2014). Optimizing $f$ to completion in the inner loop of training is computational prohibitive. Therefore, we alternate between $K$ steps of optimizing $f$ and one step of optimizing $q_\varphi$. As mentioned before, we use the multivariate Gaussian distribution with diagonal covariance matrix for the conditional $q_\phi(\boldsymbol{x}|\boldsymbol{z}) \sim \mathcal{N}(\boldsymbol{\mu}_\phi(\boldsymbol{z}), \mathrm{diag}\{\boldsymbol{\sigma}_\phi^2(\boldsymbol{z})\})$, which can be reparameterized as follows

$$\boldsymbol{x} = \boldsymbol{\mu}_\phi(\boldsymbol{z}) + \boldsymbol{\sigma}_\phi(\boldsymbol{z}) \odot \boldsymbol{\epsilon}, \quad \text{where} \quad \boldsymbol{\epsilon} \sim \mathcal{N}(0, \boldsymbol{I}),$$

where $\odot$ means the element-wise product. The corresponding score function is $\nabla_{\boldsymbol{x}} \log q_\phi(\boldsymbol{x}|\boldsymbol{z}) = -\boldsymbol{\sigma}_\phi(\boldsymbol{z})^{-1} \odot \boldsymbol{\epsilon}$. The complete procedure of SIVI-SM is formally presented in Algorithm 1.

### 3.3 THEORETICAL RESULTS REGARDING NEURAL NETWORK APPROXIMATION

The inexact lower-level optimization for neural networks introduces approximation errors. Also, neural networks themselves may introduce approximation gaps due to their approximation capacities. These numerical errors may affect the approximation quality of variational distribution $q_\varphi(\boldsymbol{x})$, which we analyze in the following proposition.

**Proposition 1.** *Let $\Omega$ be the feasible domain of $\varphi$. $\forall \varphi \in \Omega$, we say that $f_{\hat{\psi}(\varphi)}$ is $\epsilon$-accurate, if*

$$\mathbb{E}_{\boldsymbol{x} \sim q_\varphi(\boldsymbol{x})} \|\hat{R}_\varphi(\boldsymbol{x})\|_2^2 \le \epsilon, \quad \text{where} \quad \hat{R}_\varphi(\boldsymbol{x}) := S(\boldsymbol{x}) - \nabla \log q_\varphi(\boldsymbol{x}) - f_{\hat{\psi}(\varphi)}. \tag{12}$$

*Let $\varphi^*$ be one of the optimal variational parameters defined in Eq. 11 and $\tilde{\varphi}$ be the one obtained using neural network approximation defined as follows with $f_{\hat{\psi}(\varphi)}$ being $\epsilon$-accurate*

$$\tilde{\varphi} := \arg\min_{\varphi \in \Omega} \{\mathbb{E}_{\boldsymbol{x} \sim q_\varphi(\boldsymbol{x})} 2 f_{\hat{\psi}(\varphi)}(\boldsymbol{x})^T [S(\boldsymbol{x}) - \nabla_{\boldsymbol{x}} \log q_\varphi(\boldsymbol{x})] - \|f_{\hat{\psi}(\varphi)}(\boldsymbol{x})\|_2^2\}.$$

*Then we have*

$$\mathbb{E}_{\boldsymbol{x} \sim q_{\tilde{\varphi}}(\boldsymbol{x})} \|S(\boldsymbol{x}) - \nabla_{\boldsymbol{x}} \log q_{\tilde{\varphi}}(\boldsymbol{x})\|_2^2 \le \mathbb{E}_{\boldsymbol{x} \sim q_{\varphi^*}(\boldsymbol{x})} \|S(\boldsymbol{x}) - \nabla_{\boldsymbol{x}} \log q_{\varphi^*}(\boldsymbol{x})\|_2^2 + \epsilon. \tag{13}$$

See a detailed proof of Proposition 1 in Appendix C. From proposition 1, we see that the approximation error of our numerical solution to the minimax problem in 10 can be controlled by two terms.

The first term $\mathbb{E}_{\boldsymbol{x} \sim q_{\varphi^*}(\boldsymbol{x})} \| S(\boldsymbol{x}) - \nabla_{\boldsymbol{x}} \log q_{\varphi^*}(\boldsymbol{x}) \|_2^2$ measures the approximation ability of the variational distribution, and the second term measures the approximation/optimization error of the neural network $f_\psi(\boldsymbol{x})$. As long as the approximation error of $f_\psi(\boldsymbol{x})$ is small, the minimax formulation in 10 can provide variational posteriors with similar approximation accuracy to those of the original problem in 7 in terms of Fisher divergence to the target posterior.

**Remark.** When the variational parameter $\varphi$ is fixed, the lower-level optimization problem on $f_\psi(\boldsymbol{x})$ is equivalent to the following

$$\min_\psi \ \mathbb{E}_{\boldsymbol{x} \sim q_\varphi(\boldsymbol{x})} \| S(\boldsymbol{x}) - \nabla \log q_\varphi(\boldsymbol{x}) - f_\psi(\boldsymbol{x}) \|_2^2,$$

and we use this objective as a measure of approximation accuracy of $f_\psi(\boldsymbol{x})$ given $\varphi$ in Eq. 12.

## 4 EXPERIMENTS

In this section, we compare SIVI-SM to ELBO-based methods including the original SIVI and UIVI on a range of inference tasks. We first show the effectiveness of our method and illustrate the role of the auxiliary network approximation $f_\psi$ on several two-dimensional toy examples. The KL divergence from the target distributions to different variational approximations was also provided for direct comparison. We also compare the performance of SIVI-SM with both baseline methods on several Bayesian inference tasks, including a multidimensional Bayesian logistic regression problem and a high dimensional Bayesian multinomial logistic regression problem. Following Titsias & Ruiz (2019), we set the conditional layer to be $q_\phi(\boldsymbol{x}|\boldsymbol{z}) = \mathcal{N}(\boldsymbol{x}|\mu_\phi(\boldsymbol{z}), \text{diag}(\boldsymbol{\sigma}))^2$ and fix the mixing layer as $q(\boldsymbol{z}) = \mathcal{N}(0, \boldsymbol{I})$. The variational parameters therefore are $\varphi = \{\phi, \boldsymbol{\sigma}\}$. All experiments were implemented in Pytorch (Paszke et al., 2019). If not otherwise specified, we use the Adam optimizer for training (Kingma & Ba, 2014).

### 4.1 TOY EXAMPLES

We first apply SIVI-SM to approximate three synthetic distributions defined on a two-dimensional space: a banana-shaped distribution, a multimodal Gaussian, and an X-shaped mixture of Gaussian. The densities of these distributions are given in Table 2 in Appendix E. For the convenience of comparison, we used the same configuration of semi-implicit distribution family as in UIVI (Titsias & Ruiz, 2019). The $\mu_\phi(\boldsymbol{z})$ is a multilayer perceptron (MLP) with layer widths $[3, 50, 50, 2]$. The network approximation $f_\psi(\boldsymbol{x})$ is parameterized by a 4 layers MLP with layer widths $[2, 128, 128, 2]$. For SIVI-SM, we set the number of inner-loop gradient steps $K = 1$. For SIVI, we set $L = 50$ for the surrogate ELBO defined in Eq. 2. For UIVI, we used 10 iterations for every inner-loop HMC sampling. To facilitate exploration, for all methods, we used the annealing trick (Rezende & Mohamed, 2015) during training for the multimodal and X-shaped Gaussian distributions. Variational approximations from all methods were obtained after 50,000 variational parameter updates.

Figure 5 in Appendix F shows the contour plots of the synthetic distributions, together with 1000 samples from the trained variational distributions. We see that SIVI-SM produces samples that match the target distributions well. We also report the KL divergence from the target distributions to the variational posteriors (estimated via the ITE package (Szabó, 2014) using 100,000 samples from each distribution) given by different methods in Table 3 in Appendix G. We see that SIVI-SM performs better for more challenging target distributions. To better understand the role the network approximation $f_\psi(\boldsymbol{x})$ played during the training process, we visualize its training dynamics on the X-shaped distribution in Figure 1. We see that during training, $f_\psi(\boldsymbol{x})$ automatically detected where the current approximation is insufficient and guided the variational posterior towards these areas. Note the Nash-equilibrium of $f^*(\boldsymbol{x})$ in Eq. 11 is the difference between the score functions of the target distribution $p(\boldsymbol{x})$ and $q_\varphi(\boldsymbol{x})$. As the variational posterior gets closer to the target, the signal provided by $f_\psi(\boldsymbol{x})$ becomes weaker and would converge to zero in the perfect case. More details on the convergence of $f_\psi(\boldsymbol{x})$ can be found in Appendix H.

---

[2] Here $\boldsymbol{\sigma}$ is a vector with the same dimension as $\boldsymbol{x}$.

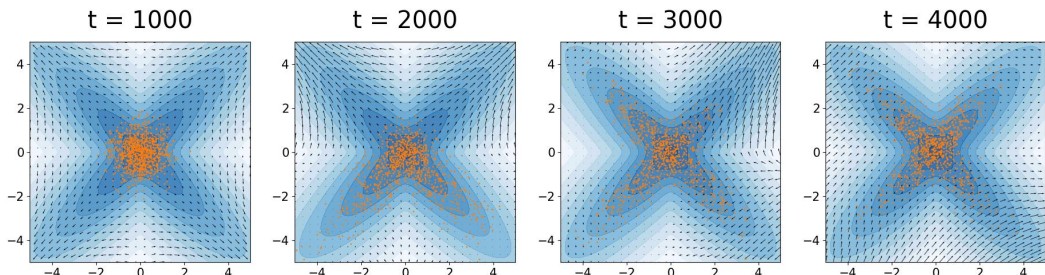

Figure 1: The quiver plots of $f_\psi(\boldsymbol{x})$ and samples from the variational posteriors during the training process on the X-shaped distribution.

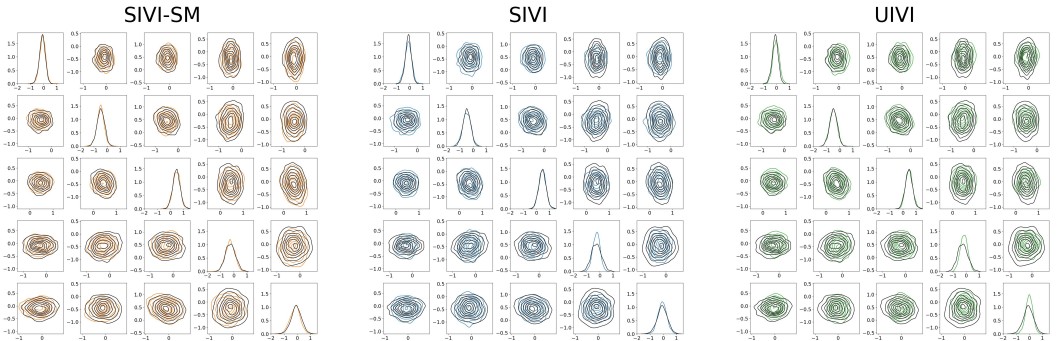

Figure 2: Comparison of the marginal and pairwise joint posteriors. The contours of the marginal and pairwise empirical densities trained by the three semi-implicit variational inference algorithms, i.e. SIVI-SM (orange), SIVI (blue) and UIVI (green), are plotted against the ground truth (black).

## 4.2 Bayesian Logistic Regression

Our second example is a Bayesian logistic regression problem where the log-likelihood function takes the following form

$$\log p(y_i|\boldsymbol{x}_i', \boldsymbol{\beta}) = y_i * \boldsymbol{\beta}^T \boldsymbol{x}_i' - \log(1 + \exp(\boldsymbol{\beta}^T \boldsymbol{x}_i')), y_i \in \{0, 1\}, \boldsymbol{x}_i' = \begin{bmatrix} 1 \\ \boldsymbol{x}_i \end{bmatrix}.$$

Here $\boldsymbol{x}_i$ are covariates, and $y_i \in \{0, 1\}$ are binary response variables. Following Yin & Zhou (2018), we set the prior as $\boldsymbol{\beta} \sim \mathcal{N}(\boldsymbol{0}, \alpha^{-1}\boldsymbol{I})$, where $\alpha = 0.01$. We consider the *waveform*[3] dataset where the dimension of $\boldsymbol{x}_i$ is 21 which leads to a parameter space of 22 dimensions. We used a standard 10-dimensional Gaussian prior for the $q(\boldsymbol{z})$. For $\mu_\phi(\boldsymbol{z})$, we used a 4 layer MLP with layer widths $[10, 100, 100, 22]$. The network approximation $f_\psi(\boldsymbol{\beta})$ is also a 4 layer MLP with layer width $[22, 256, 256, 22]$. Similarly as in section 4.1, we set the number of inner-loop gradient steps $K = 1$ in SIVI-SM. For SIVI, we set $L = 100$ and used the same training method as in Yin & Zhou (2018). For UIVI, we set the length of inner-loop HMC iterations to be 10 with the first 5 iterations discarded as burn-in, with 5 leapfrog steps in each iteration. The results of all methods were collected after 20,000 variational parameter updates.

We collected 1000 samples of $\boldsymbol{\beta}$ to represent the approximated posterior distributions for all three SIVI variants. The ground truth was formed from a long MCMC run of 400,000 iterations using parallel stochastic gradient Langevin dynamics (SGLD) (Welling & Teh, 2011) with 1000 independent particles, and a small stepsize of $10^{-4}$. Figure 2 shows the posterior estimates provided by different SIVI variants in contrast to the ground truth MCMC results. We see that SIVI and UIVI tend to slightly underestimate the variance for both univariate marginal and pairwise joint posteriors (especially for $\beta_4$, $\beta_5$), while SIVI-SM agreed with MCMC well. Furthermore, we also examined the covariance estimates of $\boldsymbol{\beta}$ and the results were presented in Figure 3. We see that SIVI-SM

---

[3]https://archive.ics.uci.edu/ml/machine-learning-databases/waveform/

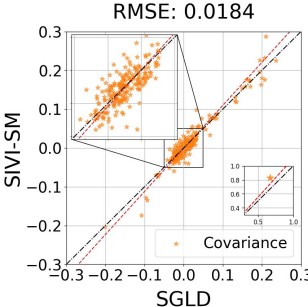 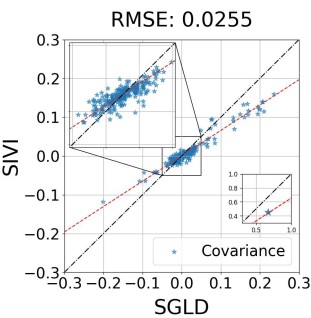 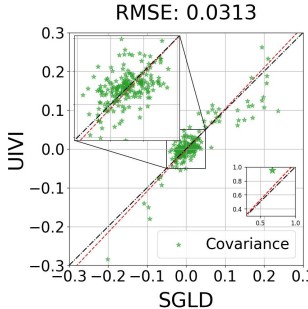

Figure 3: Scatter plot comparison of the sample covariances of the posterior. The X-axis and Y-axis represent the estimates from the ground truth MCMC runs and the corresponding SIVI variants respectively. The red lines are the regression lines.

provides the best overall approximation to the posterior which achieved the smallest rooted mean square error (RMSE) to the ground truth at 0.0184.

### 4.3 BAYESIAN MULTINOMIAL LOGISTIC REGRESSION

Our next example is a Bayesian multinomial logistic regression problem. For a data set of $N$ co-variate and label pairs $\{(x_i, y_i) : i = 1, \ldots, N\}$, where $y_i \in \{1, \ldots, R\}$, the categorical likelihood is

$$p(y_i = r | x_i) \propto \exp([1, \boldsymbol{x}_i^T] \cdot \boldsymbol{\beta}_r), \ r \in \{1, 2, \cdots, R\},$$

where $\boldsymbol{\beta} = (\boldsymbol{\beta}_1^T, \boldsymbol{\beta}_2^T, \cdots, \boldsymbol{\beta}_R^T)^T$ is the model parameter and follows a standard Gaussian prior. Following Titsias & Ruiz (2019), we used two data sets: MNIST[4] and HAPT[5]. MNIST is a commonly used dataset in machine learning that contains 60,000 training and 10,000 test instances of $28 \times 28$ images of hand-written digits which has $R = 10$ classes. HAPT (Reyes-Ortiz et al., 2016) is a human activity recognition dataset. It contains 7,767 training and 3,162 test data points, and each one of them contains features of 561-dimensional measurements captured by inertial sensors, which correspond to $R = 12$ classes of static postures, dynamic activities and postural transitions. The dimensions of the posterior distributions are 7,850 (MNIST) and 6,744 (HAPT) respectively.

We used the same variational family as before, with a 100-dimensional standard Gaussian prior for $q(\boldsymbol{z})$. We used MLPs with two hidden layers for the mean network $\mu_\phi(\boldsymbol{z})$ of the Gaussian conditional and the network approximation $f_\psi(\boldsymbol{\beta})$, with 200 hidden neurons for $\mu_\phi(\boldsymbol{z})$ and 256 hidden neurons for $f_\psi(\boldsymbol{\beta})$ for each of the hidden layers respectively. We used the same initialization of variational parameters for all methods. Following Titsias & Ruiz (2019), we used a minibatch size of 2,000 for MNIST and 863 for HAPT. As before, we set the number of inner-loop gradient steps $K = 1$ in SIVI-SM. For SIVI, we set $L = 200$ as previously done by Titsias & Ruiz (2019). For UIVI, we set the number of inner-loop HMC iterations to be 10 and discarded the first 5 iterations as burn-in, with 5 leapfrog steps in each iteration. As done in UIVI, we used the RMSProp optimizer (Tieleman & Hinton, 2012) for training. We used different batch sizes during training to investigate its effect on the quality of variational approximations for different methods. These batch sizes were selected in such a way that the corresponding computational times are comparable between different methods. See a detailed comparison on the computation times in Appendix I (the experiments were run on a RTX2080 GPU). As the gradient computation for the inner-loop HMC sampling required by UIVI is not scalable[6], the batch size for UIVI is set as $m = 1$ which was also used by Titsias & Ruiz (2019). For all methods, the results were collected after 90,000 variational parameter updates for MNIST and 40,000 variational parameter updates for HAPT.

Figure 4 shows the predictive log-likelihood on the test data as a function of the number of iterations for both data sets, where the estimates were formed based on 8,000 samples from the variational

---

[4]http://yann.lecun.com/exdb/mnist/

[5]http://archive.ics.uci.edu/ml/machine-learning-databases/00341/

[6]See a more detailed explanation in Appendix D.

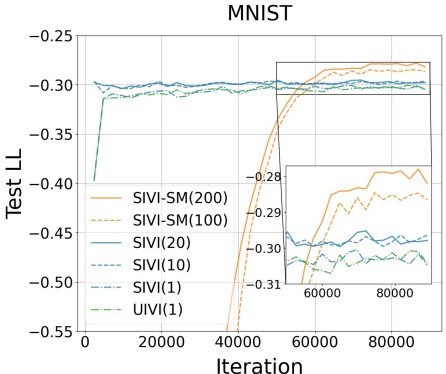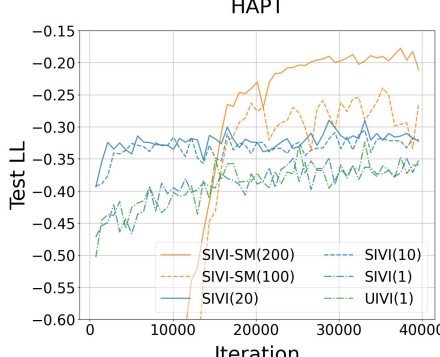

Figure 4: Estimates of the test log-likelihood for the Bayesian multinomial logistic regression model. The number in parentheses specifies the batch sizes used for training.

distributions fitted using different methods, as done in UIVI (Titsias & Ruiz, 2019). Although SIVI-SM converges slower at the beginning due to the slow training of the network approximation $f_\psi(\boldsymbol{\beta})$, it eventually surpasses the other ELBO-based variants and achieves better prediction on both datasets. Compared to ELBO-based methods, SIVI-SM would benefit more from large batch sizes.

## 4.4 BAYESIAN NEURAL NETWORKS

Lastly, we compare our method with SIVI, UIVI and SGLD on sampling the posterior of Bayesian neural network on the UCI datasets. We conduct the two-layer network with 50 hidden units and ReLU activation function. The datasets are all randomly partitioned into 90% for training and 10% for testing. We use the variational family as before, with 3-dimensional standard Gaussian prior for $q(\boldsymbol{z})$, and 10 hidden neurons for $\mu_\phi(\boldsymbol{z})$ and 16 hidden neurons for $f_\psi(\boldsymbol{x})$. The results are averaged over 10 random trials. We refer the reader to Appendix J for hyper-parameter tuning and other experiment details. Table 1 shows the average test RMSE and NLL and their standard deviation. We see that SIVI-SM can achieve on par or better results than SIVI and UIVI. Although SGLD performs better for some datasets, it requires a long run to generate samples.

Table 1: Averaged test RMSE and test negative log-likelihood of Bayesian Neural Networks on seven UCI datasets. The results were averaged from 10 independent runs.

| | AVG. TEST RMSE | | | | AVG. TEST NLL | | | |
|---|---|---|---|---|---|---|---|---|
| **Dataset** | **SIVI** | **UIVI** | **SGLD** | **SIVI-SM** | **SIVI** | **UIVI** | **SGLD** | **SIVI-SM** |
| BOSTON | $2.714_{\pm 0.08}$ | $2.817_{\pm 0.11}$ | $3.186_{\pm 0.09}$ | $\mathbf{2.621_{\pm 0.10}}$ | $2.420_{\pm 0.09}$ | $2.397_{\pm 0.07}$ | $3.164_{\pm 0.08}$ | $\mathbf{2.396_{\pm 0.13}}$ |
| CONCRETE | $6.205_{\pm 0.12}$ | $6.049_{\pm 0.10}$ | $6.512_{\pm 0.06}$ | $\mathbf{5.392_{\pm 0.09}}$ | $3.247_{\pm 0.04}$ | $3.221_{\pm 0.11}$ | $3.978_{\pm 0.05}$ | $\mathbf{3.156_{\pm 0.03}}$ |
| PROTEIN | $4.818_{\pm 0.05}$ | $4.908_{\pm 0.07}$ | $\mathbf{4.768_{\pm 0.03}}$ | $4.903_{\pm 0.04}$ | $2.994_{\pm 0.02}$ | $3.016_{\pm 0.02}$ | $\mathbf{2.979_{\pm 0.01}}$ | $3.015_{\pm 0.04}$ |
| POWER | $4.13_{\pm 0.02}$ | $4.134_{\pm 0.04}$ | $\mathbf{4.067_{\pm 0.01}}$ | $4.086_{\pm 0.05}$ | $2.853_{\pm 0.04}$ | $\mathbf{2.852_{\pm 0.06}}$ | $2.965_{\pm 0.03}$ | $2.854_{\pm 0.03}$ |
| WINEWHITE | $0.640_{\pm 0.03}$ | $0.646_{\pm 0.04}$ | $\mathbf{0.639_{\pm 0.01}}$ | $0.646_{\pm 0.06}$ | $\mathbf{0.975_{\pm 0.02}}$ | $0.983_{\pm 0.04}$ | $0.976_{\pm 0.01}$ | $0.985_{\pm 0.07}$ |
| WINERED | $0.577_{\pm 0.11}$ | $0.675_{\pm 0.06}$ | $0.651_{\pm 0.03}$ | $\mathbf{0.560_{\pm 0.08}}$ | $0.871_{\pm 0.13}$ | $1.074_{\pm 0.07}$ | $0.983_{\pm 0.03}$ | $\mathbf{0.836_{\pm 0.07}}$ |
| YACHT | $1.902_{\pm 0.21}$ | $2.1407_{\pm 0.18}$ | $2.377_{\pm 0.11}$ | $\mathbf{1.559_{\pm 0.15}}$ | $2.147_{\pm 0.11}$ | $2.309_{\pm 0.09}$ | $2.631_{\pm 0.08}$ | $\mathbf{1.683_{\pm 0.18}}$ |

## 5 CONCLUSION

We proposed SIVI-SM, a new method for semi-implicit variational inference based on an alternative training objective via score matching. Unlike the ELBO-based objectives, we showed that the score matching objective allows a minimax formulation where the hierarchical structure of semi-implicit variational families can be more efficiently exploited as the corresponding intractable variational densities can be naturally handled with denoising score matching. In experiments, we demonstrated that SIVI-SM closely matches the accuracy of MCMC in posterior estimation and outperforms two typical ELBO-based methods (SIVI and UIVI) in a variety of Bayesian inference tasks.

ACKNOWLEDGMENTS

This work was supported by National Natural Science Foundation of China (grant no. 12201014). The research of Cheng Zhang was support in part by the Key Laboratory of Mathematics and Its Applications (LMAM) and the Key Laboratory of Mathematical Economics and Quantitative Finance (LMEQF) of Peking University. The authors are grateful for the computational resources provided by the Megvii institute. The authors appreciate the anonymous ICLR reviewers for their constructive feedback.

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

## A DERIVATION OF EQ. 2-4

The gradient of ELBO is that

$$\nabla_\phi \text{ELBO} = \nabla_\phi \mathbb{E}_{q(\boldsymbol{\epsilon})q(\boldsymbol{z})} \Big[ \log p(D, \boldsymbol{x}) - \log q_\phi(\boldsymbol{x})|_{\boldsymbol{x}=T_\phi(\boldsymbol{z},\boldsymbol{\epsilon})} \Big],$$

$$= \mathbb{E}_{q(\boldsymbol{\epsilon})q(\boldsymbol{z})} \Big[ \nabla_{\boldsymbol{x}} \log p(D, \boldsymbol{x}) \nabla_\phi T_\phi(\boldsymbol{z}, \boldsymbol{\epsilon}) - \nabla_{\boldsymbol{x}} \log q_\phi(\boldsymbol{x}) \nabla_\phi T_\phi(\boldsymbol{z}, \boldsymbol{\epsilon})|_{\boldsymbol{x}=T_\phi(\boldsymbol{z},\boldsymbol{\epsilon})} \Big],$$

$$:= \mathbb{E}_{q(\boldsymbol{\epsilon})q(\boldsymbol{z})} \Big[ g_\phi^{mod}(\boldsymbol{z}, \boldsymbol{\epsilon}) - g_\phi^{ent}(\boldsymbol{z}, \boldsymbol{\epsilon}) \Big], \tag{14}$$

where

$$g_\phi^{mod}(\boldsymbol{z}, \boldsymbol{\epsilon}) := \nabla_{\boldsymbol{x}} \log p(D, \boldsymbol{x})|_{\boldsymbol{x}=T_\phi(\boldsymbol{z},\boldsymbol{\epsilon})} \nabla_\phi T_\phi(\boldsymbol{z}, \boldsymbol{\epsilon}),$$

$$g_\phi^{ent}(\boldsymbol{z}, \boldsymbol{\epsilon}) := - \nabla_{\boldsymbol{x}} \log q_\phi(\boldsymbol{x})|_{\boldsymbol{x}=T_\phi(\boldsymbol{z},\boldsymbol{\epsilon})} \nabla_\phi T_\phi(\boldsymbol{z}, \boldsymbol{\epsilon}).$$

Note that the property of margin score function,

$$\nabla_{\boldsymbol{x}} \log q_\phi(\boldsymbol{x}) = \int q_\phi(\boldsymbol{z}'|\boldsymbol{x}) \nabla_{\boldsymbol{x}} \log q_\phi(\boldsymbol{x}|\boldsymbol{z}') d\boldsymbol{z}'. \tag{15}$$

Then the gradient $g_\phi^{ent}$ can be representd as

$$g_\phi^{ent}(\boldsymbol{z}, \boldsymbol{\epsilon}) = - \mathbb{E}_{q_\phi(\boldsymbol{z}'|\boldsymbol{x})} \nabla_{\boldsymbol{x}} \log q_\phi(\boldsymbol{x}|\boldsymbol{z}')|_{\boldsymbol{x}=T_\phi(\boldsymbol{z},\boldsymbol{\epsilon})} \nabla_\phi T_\phi(\boldsymbol{z}, \boldsymbol{\epsilon}).$$

## B PROOF OF THEOREM 1

*Proof.* As discussed in section 3.1, by introducing the vector-valued function $f$, we can rewrite the optimization objective in Eq. 7 as

$$\mathbb{E}_{\boldsymbol{x} \sim q_\varphi(\boldsymbol{x})} \max_{f(\boldsymbol{x})} \{ 2f(\boldsymbol{x})^T [S(\boldsymbol{x}) - \nabla_{\boldsymbol{x}} \log q_\varphi(\boldsymbol{x})] - \|f(\boldsymbol{x})\|_2^2 \}. \tag{16}$$

Compute the score of the semi-implicit distribution $q_\varphi(\boldsymbol{x})$ defined in Eq. 1, we have

$$\nabla_{\boldsymbol{x}} \log q_\varphi(\boldsymbol{x}) = \frac{1}{q_\varphi(\boldsymbol{x})} \nabla_{\boldsymbol{x}} \int q_\xi(\boldsymbol{z}) q_\phi(\boldsymbol{x}|\boldsymbol{z}) d\boldsymbol{z}$$

$$= \frac{1}{q_\varphi(\boldsymbol{x})} \int q_\xi(\boldsymbol{z}) q_\phi(\boldsymbol{x}|\boldsymbol{z}) \nabla_{\boldsymbol{x}} \log q_\phi(\boldsymbol{x}|\boldsymbol{z}) d\boldsymbol{z}.$$

Bring the above score of $q_\varphi(\boldsymbol{x})$ into Eq. 16, we have

$$\mathbb{E}_{\boldsymbol{x} \sim q_\varphi(\boldsymbol{x})} \max_{f(\boldsymbol{x})} \{ 2f(\boldsymbol{x})^T [S(\boldsymbol{x}) - \nabla_{\boldsymbol{x}} \log q_\varphi(\boldsymbol{x})] - \|f(\boldsymbol{x})\|_2^2 \}$$

$$= \mathbb{E}_{\boldsymbol{x} \sim q_\varphi(\boldsymbol{x})} \max_{f(\boldsymbol{x})} \{ 2f(\boldsymbol{x})^T [S(\boldsymbol{x}) - \frac{1}{q_\varphi(\boldsymbol{x})} \int q_\xi(\boldsymbol{z}) q_\phi(\boldsymbol{x}|\boldsymbol{z}) \nabla_{\boldsymbol{x}} \log q_\phi(\boldsymbol{x}|\boldsymbol{z}) d\boldsymbol{z}] - \|f(\boldsymbol{x})\|_2^2 \}$$

$$= \max_{f(\boldsymbol{x})} \{ \mathbb{E}_{\boldsymbol{x} \sim q_\varphi(\boldsymbol{x})} [2f(\boldsymbol{x})^T S(\boldsymbol{x}) - \|f(\boldsymbol{x})\|_2^2] - \int q_\xi(\boldsymbol{z}) q_\phi(\boldsymbol{x}|\boldsymbol{z}) 2f(\boldsymbol{x})^T \nabla_{\boldsymbol{x}} \log q_\phi(\boldsymbol{x}|\boldsymbol{z}) d\boldsymbol{x} d\boldsymbol{z} \}$$

$$= \max_{f(\boldsymbol{x})} \{ \mathbb{E}_{\boldsymbol{z} \sim q_\xi(\boldsymbol{z}), \boldsymbol{x} \sim q_\phi(\boldsymbol{x}|\boldsymbol{z})} [2f(\boldsymbol{x})^T (S(x) - \nabla_{\boldsymbol{x}} \log q_\phi(\boldsymbol{x}|\boldsymbol{z})) - \|f(\boldsymbol{x})\|_2^2] \} \tag{17}$$

Therefor, let $\varphi = \{\xi, \phi\}$, we can rewrite the original score matching problem Eq. 7 as

$$\min_\varphi \max_f \quad \mathbb{E}_{\boldsymbol{z} \sim q_\xi(\boldsymbol{z}), \boldsymbol{x} \sim q_\phi(\boldsymbol{x}|\boldsymbol{z})} 2f(\boldsymbol{x})^T [S(\boldsymbol{x}) - \nabla_{\boldsymbol{x}} \log q_\phi(\boldsymbol{x}|\boldsymbol{z})] - \|f(\boldsymbol{x})\|_2^2.$$

If $(\varphi^*, f^*)$ defines a Nash-equilibrium of the above problem, fixing the parameters $\varphi = \varphi^*$, the optimal vector-valued function $f^*(\boldsymbol{x})$ is invariant in the derivation of Eq. 17. So we can easily deduce $f^*$ by Eq. 16

$$f^*(\boldsymbol{x}) = S(\boldsymbol{x}) - \nabla_{\boldsymbol{x}} \log q_{\varphi^*}(\boldsymbol{x}).$$

Bring $f^*(\boldsymbol{x})$ into Eq. 10, we have the unbiased approximation of $\varphi^*$

$$\varphi^* \in \arg\min_\varphi \{ \mathbb{E}_{\boldsymbol{z} \sim q_\xi(\boldsymbol{z}), \boldsymbol{x} \sim q_\phi(\boldsymbol{x}|\boldsymbol{z})} 2f^*(\boldsymbol{x})^T [S(\boldsymbol{x}) - \nabla_{\boldsymbol{x}} \log q_\phi(\boldsymbol{x}|\boldsymbol{z})] - \|f^*(\boldsymbol{x})\|_2^2 \},$$

$$\in \arg\min_\varphi \{ \mathbb{E}_{\boldsymbol{x} \sim q_\varphi(\boldsymbol{x})} \|S(\boldsymbol{x}) - \nabla_{\boldsymbol{x}} \log q_\varphi(\boldsymbol{x})\|_2^2 \}.$$

$\square$

## C  PROOF OF PROPOSITION 1

*Proof.* Consider the score matching problem with the well $\epsilon\text{-}trained$ $f_{\hat{\psi}(\varphi)}$, we have

$$
\begin{aligned}
\tilde{\varphi} &= \arg\min_{\varphi\in\Omega}\{\mathbb{E}_{\boldsymbol{x}\sim q_\varphi(\boldsymbol{x})}2f_{\hat{\psi}(\varphi)}(\boldsymbol{x})^T[S(\boldsymbol{x}) - \nabla_{\boldsymbol{x}}\log q_\varphi(\boldsymbol{x})] - \|f_{\hat{\psi}(\varphi)}(\boldsymbol{x})\|_2^2\}, \\
&= \arg\min_{\varphi\in\Omega}\{\mathbb{E}_{\boldsymbol{x}\sim q_\varphi(\boldsymbol{x})}\|S(\boldsymbol{x}) - \nabla_{\boldsymbol{x}}\log q_\varphi(\boldsymbol{x})\|_2^2 - \|S(\boldsymbol{x}) - \nabla_{\boldsymbol{x}}\log q_\varphi(\boldsymbol{x}) - f_{\hat{\psi}(\varphi)}(\boldsymbol{x})\|_2^2\}, \\
&= \arg\min_{\varphi\in\Omega}\{\mathbb{E}_{\boldsymbol{x}\sim q_\varphi(\boldsymbol{x})}\|S(\boldsymbol{x}) - \nabla_{\boldsymbol{x}}\log q_\varphi(\boldsymbol{x})\|_2^2 - \|\hat{R}_\varphi(\boldsymbol{x})\|_2^2\}.
\end{aligned} \tag{18}
$$

Therefore, we can estimate the upper bound of Fisher divergence between $S(\boldsymbol{x})$ and $\nabla_{\boldsymbol{x}}\log q_{\tilde{\varphi}}(\boldsymbol{x})$

$$
\begin{aligned}
&\mathbb{E}_{\boldsymbol{x}\sim q_{\tilde{\varphi}}(\boldsymbol{x})}\|S(\boldsymbol{x}) - \nabla_{\boldsymbol{x}}\log q_{\tilde{\varphi}}(\boldsymbol{x})\|_2^2, \\
=&\mathbb{E}_{\boldsymbol{x}\sim q_{\tilde{\varphi}}(\boldsymbol{x})}\|S(\boldsymbol{x}) - \nabla_{\boldsymbol{x}}\log q_{\tilde{\varphi}}(\boldsymbol{x})\|_2^2 - \|\hat{R}_{\tilde{\varphi}}(\boldsymbol{x})\|_2^2 + \|\hat{R}_{\tilde{\varphi}}(\boldsymbol{x})\|_2^2, \\
\leq&\mathbb{E}_{\boldsymbol{x}\sim q_{\varphi^*}(\boldsymbol{x})}\|S(\boldsymbol{x}) - \nabla_{\boldsymbol{x}}\log q_{\varphi^*}(\boldsymbol{x})\|_2^2 - \|\hat{R}_{\varphi^*}(\boldsymbol{x})\|_2^2 + \|\hat{R}_{\tilde{\varphi}}(\boldsymbol{x})\|_2^2, \\
\leq&\mathbb{E}_{\boldsymbol{x}\sim q_{\varphi^*}(\boldsymbol{x})}\|S(\boldsymbol{x}) - \nabla_{\boldsymbol{x}}\log q_{\varphi^*}(\boldsymbol{x})\|_2^2 + \epsilon,
\end{aligned} \tag{19}
$$

where $\varphi^* := \arg\min_{\varphi\in\Omega}\{\mathbb{E}_{\boldsymbol{x}\sim q_\varphi(\boldsymbol{x})}\|S(\boldsymbol{x}) - \nabla_{\boldsymbol{x}}\log q_\varphi(\boldsymbol{x})\|_2^2\}$. And the last inequality is due to the fact that $f_{\hat{\psi}(\varphi)}$ is well $\epsilon\text{-}trained$ and $\|\hat{R}_{\varphi^*}(\boldsymbol{x})\|_2^2$ is non-negative. $\square$

## D  COMPUTATIONAL ISSUES ON UIVI

Unlike SIVI that samples from the prior $q(\boldsymbol{z})$, UIVI samples from the posterior distribution $q(\boldsymbol{z}|\boldsymbol{x})$, which can provide unbiased gradient estimate to the exact ELBO (Titsias & Ruiz, 2019). However, UIVI requires computing the gradient of $\log q(\boldsymbol{z}|\boldsymbol{x})$ during the iterations of HMC sampling procedures. If UIVI uses a minibatch of $m$ data points $\boldsymbol{x}_1, \boldsymbol{x}_2, \cdots, \boldsymbol{x}_m$ in the training process, it needs to compute the Jacobian matrix $[\nabla_{\boldsymbol{z}}\log q(\boldsymbol{z}|\boldsymbol{x}_1), \nabla_{\boldsymbol{z}}\log q(\boldsymbol{z}|\boldsymbol{x}_2), \cdots, \nabla_{\boldsymbol{z}}\log q(\boldsymbol{z}|\boldsymbol{x}_m)]$, which is not scalable for automatic differentiation using backpropagation. Therefore, we set the batch size $m = 1$ for UIVI as done in Titsias & Ruiz (2019).

## E  DENSITIES FOR THE TOY EXAMPLES

Table 2: Synthetic target distributions used in the toy experiments.

| Name | $p(\boldsymbol{x})$ | Parameters |
|---|---|---|
| Banana-shaped | $\boldsymbol{x} = (v_1, v_1^2 + v_2 + 1), \boldsymbol{v} \sim \mathcal{N}(\boldsymbol{0}, \Sigma)$ | $\Sigma = \left[\begin{smallmatrix} 1 & 0.9 \\ 0.9 & 1 \end{smallmatrix}\right]$ |
| Multimodal | $\frac{1}{2}\mathcal{N}(\boldsymbol{x}|\boldsymbol{\mu}_1, \boldsymbol{I}) + \frac{1}{2}\mathcal{N}(\boldsymbol{x}|\boldsymbol{\mu}_2, \boldsymbol{I})$ | $-\boldsymbol{\mu}_1 = \boldsymbol{\mu}_2 = [2, 0]^T$ |
| X-shaped | $\frac{1}{2}\mathcal{N}(\boldsymbol{x}|\boldsymbol{0}, \Sigma_1) + \frac{1}{2}\mathcal{N}(\boldsymbol{x}|\boldsymbol{0}, \Sigma_2)$ | $\Sigma_1 = \left[\begin{smallmatrix} 2 & 1.8 \\ 1.8 & 2 \end{smallmatrix}\right], \Sigma_2 = \left[\begin{smallmatrix} 2 & -1.8 \\ -1.8 & 2 \end{smallmatrix}\right]$ |

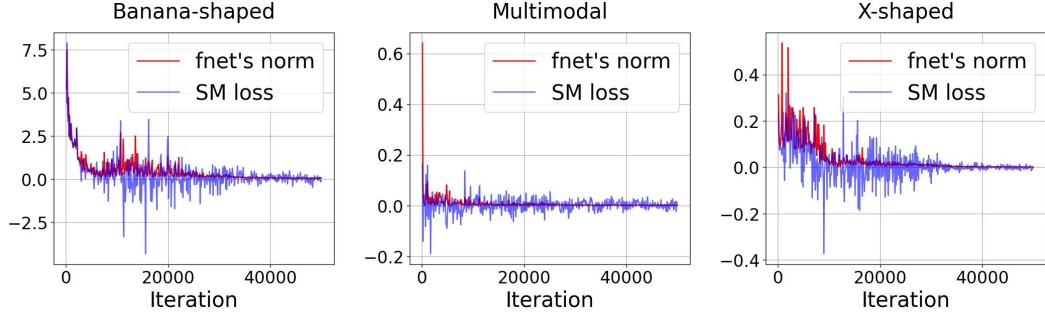

Figure 6: The training loss and $L_2$-norm of $f_\psi(\boldsymbol{x})$. $\mathbb{E}_{q_\varphi(\boldsymbol{x})}\|f_\psi(\boldsymbol{x})\|_2^2$ (**fnet's norm**) and the training objective loss (**SM loss**) in Eq. 10 are both estimated using 500 samples.

## F  SIVI SAMPLES ON THE TOY EXAMPLES

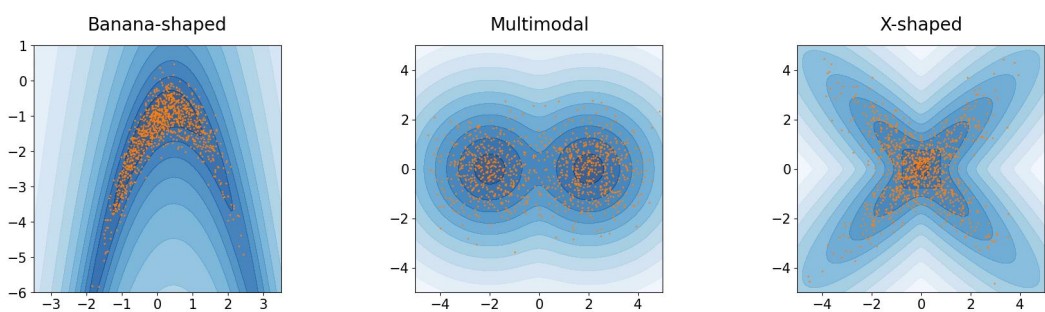

Figure 5: Performance on toy examples. The samples from the variational posteriors fitted with SIVI-SM (orange) match the shape of target distributions (blue).

## G  KL DIVERGENCE RESULTS ON THE TOY EXAMPLES

Table 3: KL divergence from the target to the variational posteriors. The results were averaged from 5 independent runs with one standard deviation in the parentheses.

| Name | SIVI | UIVI | SIVI-SM(ours) |
|---|---|---|---|
| Banana-shaped | **0.1876(0.0230)** | 0.3602(0.1106) | 0.1936(0.0169) |
| Multimodal | 0.1823(0.0025) | 0.0611(0.0192) | **0.0005(0.0007)** |
| X-shaped | 0.0341(0.0118) | 0.0236(0.0083) | **0.0046(0.0038)** |

## H  CONVERGENCE PERFORMANCE OF SIVI-SM

Here, we demonstrate the convergence behavior of SIVI-SM in our experiments. For the topy examples in section 4.1, we use $500$ samples from $q_\varphi(\boldsymbol{x})$ to form the Monte Carlo estimates of the loss (**SM loss**) in Eq. 10, and the $L_2$-norm $\mathbb{E}_{q_\varphi(\boldsymbol{x})}\|f_\psi(\boldsymbol{x})\|_2^2$ of the $f_\psi$ function(**fnet's norm**) during the training process. Figure 6 shows the estimated **SM loss** and **fnet's norm** as a function of the number of iterations for the three synthetic toy distributions.

Similarly, Figure 7 shows the convergence traces of the **SM loss** and **fnet's norm** in the experiments in section 4.3. Note that although the dimensions of the posterior distributions are high, i.e. 7850 for MNIST and 6744 for HAPT, the corresponding **fnet's norms** can be quite low ($58.510$ for MNIST

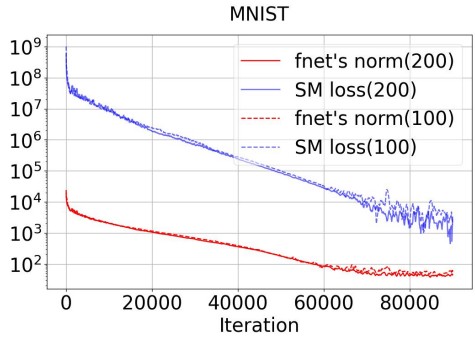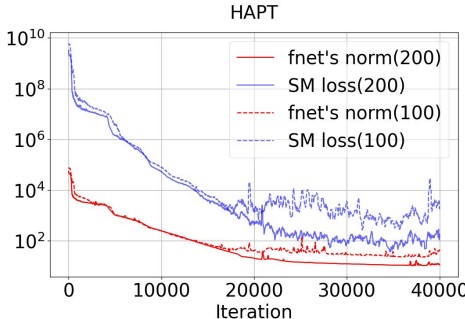

Figure 7: Loss convergence for MNIST and HAPT. The loss trace has been smoothed with a rolling window of size 5.

and $8.267$ for HAPT), indicating the effectiveness of our methods even in high dimensional spaces.

## I   SECONDS PER ITERATION IN FIGURE 4

The following table shows the run times of different methods per iteration on a RTX2080 GPU. We see that the run times for SIVI and SIVI-SM are comparable with the chosen pairs of batch sizes (i.e., 10 vs 100 and 20 vs 200). As discussed before, the inner-loop HMC iterations make UIVI slower than other methods.

Table 4: Seconds per iteration for MNIST and HAPT.

| MNIST | | | | HAPT | | | |
|---|---|---|---|---|---|---|---|
| **Method** | **S/it** | **Method** | **S/it** | **Method** | **S/it** | **Method** | **S/it** |
| SIVI(20) | 0.0107 | SIVI-SM(200) | 0.0088 | SIVI(20) | 0.0048 | SIVI-SM(200) | 0.0059 |
| SIVI(10) | 0.0065 | SIVI-SM(100) | 0.0058 | SIVI(10) | 0.0045 | SIVI-SM(100) | 0.0057 |
| SIVI(1) | 0.0042 | UIVI(1) | 0.0507 | SIVI(1) | 0.0042 | UIVI(1) | 0.0493 |

## J   EXPERIMENT SETTING FOR BAYESIAN NEURAL NETWORKS

For SIVI, we set $L = 100$ the batch size is $m = 10$ in training process. For UIVI, the setting of HMC inner loop is similar with section 4.3. For SGLD, we choose the step size from $\{10^{-4}, 10^{-5}, 10^{-6}\}$ and iteration number in $\{50000, 100000\}$ by validation in training process with 100 particles. For SIVI-SM, we set inner-loop gradient steps $K = 1, 3$ by validation and run 20,000 iterations for training.

## K   RELATED METHODS

Consider a test functions class $\mathcal{F}$, the Stein discrepancy (Gorham & Mackey, 2015) measure between $p$ and $q$ is defined follows

$$\mathbb{S}(q, p) = \sup_{f \in \mathcal{F}} \mathbb{E}_{q(\boldsymbol{x})} \left[ \nabla_{\boldsymbol{x}} \log p(\boldsymbol{x})^T f(\boldsymbol{x}) + \text{Tr}(\nabla_{\boldsymbol{x}} f(\boldsymbol{x})) \right]. \tag{20}$$

This measure is based on the following Stein's identity (Stein, 1972)

$$\mathbb{E}_{q(\boldsymbol{x})} \left[ \nabla_{\boldsymbol{x}} \log q(\boldsymbol{x})^T f(\boldsymbol{x}) + \text{Tr}(\nabla_{\boldsymbol{x}} f(\boldsymbol{x})) \right] = 0. \tag{21}$$

An early example in variational inference used Stein discrepancy is operator variational inference (OPVI) (Ranganath et al., 2016), which constructs a variational operator (e.g. Langevin-Stein Op-

erator $O_{\text{LS}}^p$) objective [7]

$$\mathcal{L}(q, O_{\text{LS}}^p, \mathcal{F}) = \sup_{f \in \mathcal{F}} \left( \mathbb{E}_{q(\boldsymbol{x})} \left[ \nabla_{\boldsymbol{x}} \log p(\boldsymbol{x})^T f(\boldsymbol{x}) + \text{Tr}(\nabla_{\boldsymbol{x}} f(\boldsymbol{x})) \right] \right)^2.$$

Then OPVI solves the minmax optimization problem simultaneously with $q$ and $f$. Unlike OPVI, learned Stein discrepancy (LSD) (Grathwohl et al., 2020) utilizes the $L_2$ regularization term to substitute the constraint of $\mathcal{F}$

$$\mathcal{L}_{\text{LSD}} = \sup_f \mathbb{E}_{q(\boldsymbol{x})} \left[ \nabla_{\boldsymbol{x}} \log p(\boldsymbol{x})^T f(\boldsymbol{x}) + \text{Tr}(\nabla_{\boldsymbol{x}} f(\boldsymbol{x})) - \lambda \|f(\boldsymbol{x})\|_2^2 \right]. \tag{22}$$

In fact, the variational objective of SIVI-SM in Eq.8 can be viewed as $\mathcal{L}_{\text{LSD}}$. Bring Eq.21 into Eq.22 and let $\lambda = \frac{1}{2}$, we have

$$\mathcal{L}_{\text{LSD}} = \sup_f \mathbb{E}_{q(\boldsymbol{x})} \left[ f(\boldsymbol{x})^T \left( \nabla_{\boldsymbol{x}} \log p(\boldsymbol{x}) - \nabla_{\boldsymbol{x}} \log q(\boldsymbol{x}) \right) - \frac{1}{2} \|f(\boldsymbol{x})\|_2^2 \right],$$

$$= \frac{1}{2} \mathbb{E}_{q(\boldsymbol{x})} \| \nabla \log p(\boldsymbol{x}) - \nabla \log q(\boldsymbol{x}) \|_2^2.$$

However, OPVI and LSD both involve the $\text{Tr}(\nabla_{\boldsymbol{x}} f(\boldsymbol{x}))$ term which is not easy to compute for high dimensional problems. Our method takes a further step by utilizing the hierarchical structure of $q(\boldsymbol{x})$ in Eq.1. Using a mathematical trick that is similar to denoising score matching, we arrive at a formulation that easily scales up to high dimensions.

## L ON THE BIASENESS OF THE MC GRADIENT ESTIMATE OF THE FISHER DIVERGENCE IN EQ. 7

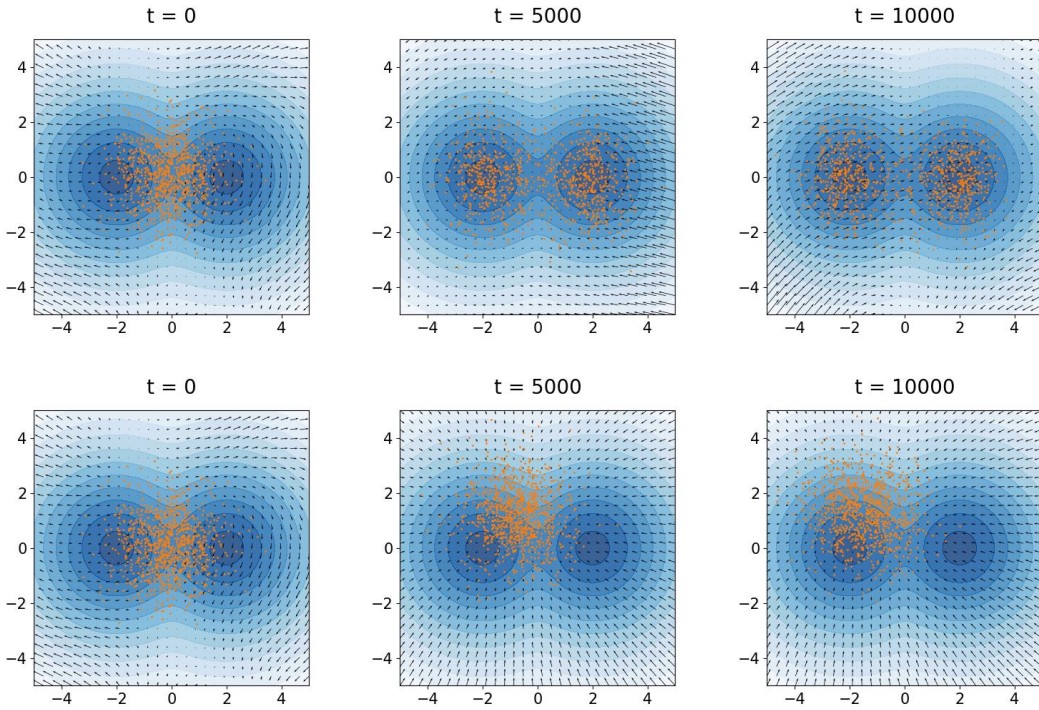

Figure 8: The quiver plots of $f_\psi(\boldsymbol{x})$ and samples from the variational posteriors during the training process of the multimodal Gaussian distribution. **Up:** SIVI-SM. **Bottom:** Biased gradient estimates.

---

[7]The objective is similar with the definition of Stein measure in (Liu et al., 2016).

