# OpenReview forum: "Semi-Implicit Variational Inference via Score Matching"
_ICLR.cc/2023/Conference — ICLR 2023 notable top 25%_

### Official Review · Reviewer_sufy · 2022-10-24

**Confidence:** 5
**Correctness:** 3
**Technical Novelty And Significance:** 3
**Empirical Novelty And Significance:** 2
**Recommendation:** 6

**Clarity, Quality, Novelty And Reproducibility:**


The manuscript is presented very clearly and the writing is of high quality (definitely above the average ML conference paper). In particular, as someone who is quite familiar with the literature, I found the authors did a superb job with the introductory and background sections up to and including Page 3, which were very well-written, comprehensive, and succinctly summarizes the narrative of this line of work up to the current point. The exposition of the proposed method was also quite clear, of high technical quality and substantiated by proofs where necessary. The experiments were conceived and executed well, the descriptions were clear and included the details necessary for reproduction, and the results are presented nicely (I especially appreciated the qualitative examples and visualizing the role of the auxiliary network in the quiver plot). The only thing I would say is that the problems considered are all fairly toy by today's standards. I would've liked to see more challenging problems being tackled by the proposed method.

There is no mention of source code availability, nor indication it will become available upon publication. This is will be crucial for reproducibility, particularly for methods that are notoriously finicky (minimax optimization problems).

The novelty is clear. This provides an interesting and practical technique for solving the Fisher divergence minimization problem by reducing it to another optimization problem and leveraging existing neat tricks such as denoising score matching.

**Strength And Weaknesses:**


The clarity of the writing and the soundness of the proposed technique are definite strengths of this work, as is the novelty. These are discussed in further detail in the subsequent section.

One significant weakness I wish to highlight is the conceptual motivation behind this approach at a higher level. Presumably, the appeal of SIVI-type methods is that one is able to circumvent the need to solve these minimax/saddle point optimization problems, the instability and difficulty of which have long plagued GANs and their derivative methods. SIVI-type enables one to steer clear of this problematic paradigm altogether, but the proposed method brings us right back to where we started. Although the experimental results indicate benefits over traditional SIVI-type methods, it's difficult to assess how this approach will generalize beyond the three relatively simple problems considered here. I'm interested to hear what the authors have to say and would like to see this discussed in the manuscript. Is there something about the formulation (Fisher divergence, score matching objective, denoising) that fundamentally changes the problematic nature of this minimax game?

Secondly, at a more fundamental level, this paper adopts an unconventional paradigm of variational inference, namely, minimizing the Fisher divergence. This is not a well-studied case (unlike other divergences in the f-divergence family beyond the reverse KL divergence, such as the alpha-divergence). The only other case of this that I've seen is Yang et al. 2019 (https://arxiv.org/abs/1905.05284) which has been cited in this manuscript but is not widely recognized by the community (has it been accepted by a peer-reviewed publication yet?) This facet should be given more attention. Particularly, in this sense, it is difficult to directly draw comparisons between SIVI/UIVI and this method, since it is solving a fundamentally different problem. To what extent are the improvements derived from using this different objective? It seems this manuscript could benefit from some ablation studies.

### Misc questions and issues


Regarding footnote 2, why not just write the covariance as the scale-identity matrix "\sigma \mathbf{I}"?

In figure 3, in the SIVI-SM pane, the colour of the true and approximate densities are too similar (red and orange, respectively) which makes it far more difficult to discern their differences.  In contrast, the colours representing the other methods in the other panes are quite different to that of the ground truth, which makes the presentation slightly unfair.

In Section 4.2, regarding the covariance estimates of \beta (shown in Figure 4) is this simply \sigma? Please clarify.

Typos:
- "log-likeilhood"
- "in term of"
- "get ride of"
- "ground truth _were_ formed"
- "all _the_ three SIVI variants"
- number of missing commas all throughout the text. The manuscript could generally benefit from a few more passes of copy-editing.


**Summary Of The Paper:**

This paper proposes a new approach to semi-implicit variational inference (SIVI). SIVI-type formulations greatly increase the expressiveness of variational posteriors but introduce intractabilities to their inference. Existing methods rely on additional lower bounds on the evidence lower bound (ELBO), or require an expensive Markov chain Monte Carlo (MCMC) estimate of an inner gradient expectation. This work proposes the use of an alternative objective (the Fisher divergence), which, combined with the hierarchical nature of semi-implicit variational distributions can be approximated as the solution (Nash equilibrium) of saddle point optimization problem. This insight leverages the established link between denoising and score matching.

**Summary Of The Review:**

While I am not entirely convinced about some fundamental aspects of this work (as discussed in "Weaknesses") I am leaning toward recommending acceptance of this paper. I am looking forward to discussions of the points raised.

---

> ### Author Response · Authors · 2022-11-19
> **Response to Reviewer sufy**
>
> Thank you for your thoughtful review and valuable feedback. We address your specific questions and comments below
>
> Q1. Although the experimental results indicate benefits over traditional SIVI-type methods, it's difficult to assess how this approach will generalize beyond the three relatively simple problems considered here. I'm interested to hear what the authors have to say and would like to see this discussed in the manuscript. Is there something about the formulation (Fisher divergence, score matching objective, denoising) that fundamentally changes the problematic nature of this minimax game?
>
> A1. Regarding the complexity of the problems in the experiments which is also mentioned by the other reviewers, we now have added an experiment on Bayesian neural networks in section 4.4, which have complicated multi-modal posteriors. The results show that our method performs on par or better that the other SIVI variants. For the minimax formulation, we want to clarify two advantages against the minimax game in GAN training. First, our minimax formulation does not involve samples from two distributions. One difficulty for GAN training is that when the two distributions are far away from each other, the discriminator (trained using samples from the two distributions) can not provide enough information for the generator. In our case, we only have to estimate the score of the variational posterior, and the score of the target distribution is available. Moreover, density ratio estimation is known to be challenging in high dimensions (Sugiyama et al., 2012) and when used for VI (i.e., Mescheder et al., 2017) one also needs to choose a reference distribution which would be tricky (Mescheder et al proposed an adaptive contrast strategy in their AVB paper). On the other hand, score estimation via denoising score matching is more stable, and has been widely used in diffusion models. Secondly, using the denoising score matching trick, we maintain the dependency of the score of the variational posterior on the variational parameter $\varphi$ in the objective function. While the discriminator in GAN breaks the dependency of the density ratio on the parameter of the generator, and restoring this dependency could be helpful (Metz et al 2017, Unrolled GAN paper).
>
>
> Q2. Particularly, in this sense, it is difficult to directly draw comparisons between SIVI/UIVI and this method, since it is solving a fundamentally different problem. To what extent are the improvements derived from using this different objective? It seems this manuscript could benefit from some ablation studies.
>
> A2. We agree that it is hard to directly compare ELBO-based methods and ours. In the experiments, we compare our methods with these ELBO-based alternatives in terms of the quality of posterior estimation. For all methods, the semi-implict variational family is the same and the only difference is the training objective. From a computational perspective, the main benefit of using Fisher divergence objective is that we do not use surrogate ELBOs nor require inner-loop HMC iterations. We did some ablation study on the mini-batch sizes in section 4.3. A deeper theoretical understanding of the difference between ELBO and Fisher divergence for VI would be an interesting topic for future work.
>
> Q3. Regarding footnote 2, why not just write the covariance as the scale-identity matrix "$\sigma \mathbf{I}$"? In Section 4.2, regarding the covariance estimates of $\beta$ (shown in Figure 4) is this simply $\sigma$? Please clarify.
>
> A3. Thanks for careful reading! The $\sigma$ there is a vector, so it would lead to a diagonal matrix with potentially different diagonal elements (we have a footnote explanation of this). The covariance estimates of $\beta$ in section 4.2 are for the posterior distribution, so it is not the $\sigma$ here.
>
> Q4. The only thing I would say is that the problems considered are all fairly toy by today's standards. I would've liked to see more challenging problems being tackled by the proposed method.
>
> A4. We have added an experiment on Bayesian neural networks in section 4.4, which have complicated multi-modal posteriors.
>
> Q5. There is no mention of source code availability, nor indication it will become available upon publication. This is will be crucial for reproducibility, particularly for methods that are notoriously finicky (minimax optimization problems).
>
> A5. We will clean the code soon and make it available upon publication.
>
> Thanks for catching the typos and the suggestion for figure presentation! We have corrected these typos and incorporated your suggestions in our revision.

---

> > ### Comment · Reviewer_sufy · 2022-12-03
> > **Ack**
> >
> > Thanks for your response. Firstly, I think the addition of the experiment on BNNs goes a long way toward addressing the lack of a more challenging experimental problem studied in this paper, which seems to be a key concern shared amongst all reviewers. Secondly, I found that Reviewer cW7H raised an interesting point on the connections to the Stein discrepancy, which has now been studied extensively in variational inference. The authors' response has addressed seems to have addressed some of their concerns. However, for a more complete and rigorous empirical analysis, it would be beneficial to benchmark against one or more of these methods as baselines. In particular, it would be compelling to see the proposed method excel on high-dimensional problems in which traditional Stein-discrepancy-based methods are purported to struggle. Overall, I am not too inclined to adjust my score, as while some improvements have been made, further albeit minor shortcomings have come to light.
> >
> > Re: A3. Thanks for the clarification. I think I was confused because \sigma sometimes appears in boldface and sometimes not (e.g. in the very next line) so if you could fix this that would make it less ambiguous.

---

> > > ### Author Response · Authors · 2022-12-04
> > > **Thanks for the Ack**
> > >
> > > Thanks for your acknowledgement. **Regarding stein discrepancy based variational inferece, we want to clarify that those methods are mainly particle-based methods that can only provide finite (which is the number of particles) samples for each run. If new samples are required, one needs to retrain the particles which would be costly.** In this sense, particle-based VI methods is more like MCMC. On the other hand, semi-implicit VI is more like typical VI methods than can generate new samples instantaneously once trained. **Moreover, the original SIVI paper (Yin and Zhou, 2018) indeed has provided a comparison to SVGD on a Bayesian logistic regression model. Their results show that SVGD may perform poorly for posterior estimation which leads to misrepresented uncertainty.**
> > >
> > > Re:Re: A3. Thanks again for detecting this ambiguous notation issue. We will modify accordingly in our revision.

---

### Official Review · Reviewer_egpL · 2022-10-24

**Confidence:** 3
**Correctness:** 4
**Technical Novelty And Significance:** 3
**Empirical Novelty And Significance:** 2
**Recommendation:** 8

**Clarity, Quality, Novelty And Reproducibility:**

**Clarity**

I found this paper to be clear. I have some minor suggestions
* The first paragraph ends (and the second starts) in a slightly strange position. Since the second paragraph talks about the limitations of the factorised approximation, I would suggest introducing this approximation here instead of at the end of the first paragraph.
* I would suggest that implicit and semi-implicit distributions are defined more explicitly (no pun intended) in the introduction, to aid readers that don't already know what they are.
* K appearing in the SIVI bound is not defined as far as I can tell
* I did not find the connection that the authors make to demonising score matching to be particularly clear since the proposed method does not perturb the variable of interest. Perhaps the authors can give some more thought to how to explain this connection.
* I would suggest the authors include notation that places more focus on the cancellation of the variational density in the denominator of equation 9. Perhaps this can be done by including it crossed out in the numerator and denominator. This is a very simple step, but I was left confused by it for a bit because of how the prose reads.
* In algorithm 1 the notation for the L2 norm (i.e. $|| \cdot ||_{2}^2$) is missing for the regularisation of the surrogate function.
* I did not understand what was being plotted in figure 4 after reading both the caption and the prose that describes it.

**Novelty**

* To the best of my knowledge the proposed method is novel and non-trivial

**Reproducibility**

* The authors do not provide code but I think that the provided experimental detail is enough for me to be able to reproduce the experiments.


**Strength And Weaknesses:**

**Strengths**

* The paper presents a neat idea that makes sense and seems to work reasonably well
* The proposed method is clear and easy to follow.


**Weaknesses**


 The experiments are relatively simple and do not cover a very diverse range of cases.
* All of the non-toy posteriors considered are convex. This does not allow the reader to get an idea of how well the proposed method would fare under multimodal posteriors.

* The UIVI paper validates their approach using VAEs. These models can present a multimodal posterior distribution over the latent space. Perhaps it could be good to replicate this experiment.


* The authors motivate their method by observing the biasedness of the MC estimate of the Fisher divergence in equation 7. I think that a small experiment showing the unsuitability of this estimator (even in the appendix if there is no space for it in the main text) would make the argument much more convincing.

* As a practitioner, I always struggle to imagine in which types of situations it would be best to apply a more exotic inference method, like the one described in this paper, instead of HMC. I would suggest that the authors include HMC with a compute allowance matching that of the implicit VI methods as a baseline.



**Summary Of The Paper:**

This paper proposes a novel score-matching-based method to learn semi-implicit variational distributions. In semi-implicit models, the score of the variational distribution is challenging to estimate directly and Monte Carlo estimates of the score matching objective are biased due to the loss being a non-linear function of the scores. The authors instead re-write the squared distance between the score of the distribution being approximated and the score of the variational distribution as the inner product of the score difference with the output of a learnable function. The authors put forth a ridge regression style learning objective for this auxiliary function which is optimised when the function matches the score difference and thus the exact squared distance between scores is recovered. At the same time, this expression is now only linear in the score of the variational distribution, allowing for learning with unbiased MonteCarlo estimates of the gradients of variational parameters. This results in a min-max objective, where the auxiliary function is trained to approximate the error in the variational distribution’s score and the variational distribution is optimised such that its score matches that of the desired distribution.

The authors validate their method on some toy 2d densities, 22-dimensional logistic regression and multinomial logistic regression on the MNIST (7850 dimensions) and HAPT (6744 dimensions) datasets. As baselines, Semi-implicit variational inference (SIVI) and unbiased-implicit variational inference (UIVI) are used. The proposed method performs on par or better than baselines on all tasks. Another advantage of the proposed method is that it admits minibatching, unlike UIVI, which relies on HMC updates.

**Summary Of The Review:**

I like this paper and would like to see it accepted at ICLR. However, in their current state, the experiments seem to be a bit weak for what I would expect from an ICLR paper. If the authors address the issues described above in this regard I would be happy to raise my score to an 8.

---

> ### Author Response · Authors · 2022-11-19
> **Response to Reviewer egpL**
>
> Thank you for your thoughtful review and valuable feedback. We address your specific questions and comments below
>
> Q1. All of the non-toy posteriors considered are convex. This does not allow the reader to get an idea of how well the proposed method would fare under multimodal posteriors. The UIVI paper validates their approach using VAEs. These models can present a multimodal posterior distribution over the latent space. Perhaps it could be good to replicate this experiment.
>
> A1. Thanks for your suggestion! SIVI and UIVI are ELBO-based methods that naturally adapt to VAE models. Since our method uses fisher divergence as the training objective, it can not be directly applied to VAEs. We did an additional experiment on Bayesian neural networks in section 4.4, which also have multimodal posteriors.
>
> Q2. The authors motivate their method by observing the biaseness of the MC estimate of the Fisher divergence in equation 7. I think that a small experiment showing the unsuitability of this estimator (even in the appendix if there is no space for it in the main text) would make the argument much more convincing.
>
> A2. We added an experiment on the toy mixture of two Gaussians example to Appendix J, which demonstrates the unsuitablity of the biased gradient estimator.
>
> Q3. As a practitioner, I always struggle to imagine in which types of situations it would be best to apply a more exotic inference method, like the one described in this paper, instead of HMC. I would suggest that the authors include HMC with a compute allowance matching that of the implicit VI methods as a baseline.
>
> A3. First, the main advantage of variational inference(VI) against MCMC is that the sampling process is fast (no further iterations needed). While classical VI methods (e.g., mean field VI) often fail to provide accurate posterior approximation, semi-implicit VI methods discussed in this paper enriches the expressiveness of the variational family and can provide posterior approximations that are comparable to MCMC. We believe that when it is necessary to sample the target distribution many times or quickly generate samples, VI with more expressive variational distributions is a better choice than MCMC.
>
> Q4. I did not find the connection that the authors make to denoising score matching to be particularly clear since the proposed method does not perturb the variable of interest. Perhaps the authors can give some more thought to how to explain this connection.
>
> A4. You are right that SIVI-SM does not perturb the target distribution. In fact, the connection is that the semi-implicit variational family $q(x) = \int q(x|z)q(z)dz$ can be viewed as a perturbed distribution where $q(z)$ is the prior distribution and $q(x|z)$ is the perturbation. It is this latent hierarchical structure that enables our reformulation and denoising score matching.
>
> Q5. I did not understand what was being plotted in figure 4 after reading both the caption and the prose that describes it.
>
> A5. Figure 4 shows the scatter plots of covariance estimates given by different SIVI variants in contrast to SGLD. For example, in the first plot of figure 4, we use the 1000 samples (each being a 22-dimensional vector) from SIVI-SM and SGLD respectively to estimated the covariances $\mathbb{C}\mathrm{ov}(x_i,x_j)$ for $1\leq i\leq j\leq 22$. Then we construct a scatter plot of $(c_{i,j}^{\text{sivi-sm}}, c_{i,j}^{\text{sgld}}) $ for all $1\leq i\leq j\leq 22$, where the $c_{i,j}^{\text{sivi-sm}}$  and $c_{i,j}^{\text{sgld}}$ are the estimates of $\mathbb{C}\mathrm{ov}(x_i,x_j)$ by SIVI-SM and SGLD. The SGLD samples were obtained from a long run with a small stepsize, which we take as the ground truth. And if the SIVI-SM method approximates the covariance well, the scatters will be close to the diagonal. The red lines are the regression lines for reference.
>
> Other clarity issues are addressed in our revision. Thanks for your suggestions!

---

> > ### Comment · Reviewer_egpL · 2022-11-20
> > **Thanks for the reply.**
> >
> > Thanks for the detailed reply. I think the new experiments make the paper much stronger and I have updated my score accordingly.
> >
> > On a different note, it is possible to apply sampling-based inference to VAEs. See for example https://arxiv.org/pdf/2202.04599.pdf

---

> > > ### Author Response · Authors · 2022-11-27
> > > **Reply**
> > >
> > > We would like to thank you again for your effort and positive feedback! Your endorsement is very important to us. We are very happy that our response and updated manuscript have resolved your questions. Your valuable comments help us a lot and make our paper stronger.
> > >
> > > Thanks for sharing the interesting paper on sampling-based inference to VAEs! We'll read it carefully to see how our method can be adapted to hierarchical VAEs.
> > >
> > > Cheers,
> > >
> > > Authors

---

### Official Review · Reviewer_cW7H · 2022-10-26

**Confidence:** 4
**Correctness:** 3
**Technical Novelty And Significance:** 3
**Empirical Novelty And Significance:** 2
**Recommendation:** 6

**Clarity, Quality, Novelty And Reproducibility:**

This can be a solid and original paper if the weaknesses are sufficiently addressed. The clarify is good but can be improved. One place I found unclear is

* Section 3.1, "Although the hierarchical structure of q(x) allows us to estimate its score function via denoising score matching, the estimated score function would break the dependency ...". I don't understand this. Can you clarify?

**Strength And Weaknesses:**

## Strengths

* The idea of rewriting score matching loss using its dual norm is an interesting one---it leads to a nice form that does not require computing the density/score of the hierarchical variational posterior.

* The authors provide theoretical justification of not solving the inner max problem to the optimum.

## Weaknesses

* Although directly using score matching loss as the variational objective is relatively under explored, its close relatives---Stein discrepancies----have been studied extensively.

There is a known connection between score matching loss and Stein discrepancies, see, e.g.,
Liu, Q., Lee, J., & Jordan, M. (2016, June). A kernelized Stein discrepancy for goodness-of-fit tests. In International conference on machine learning (pp. 276-284). PMLR.
Barp, A., Briol, F. X., Duncan, A., Girolami, M., & Mackey, L. (2019). Minimum stein discrepancy estimators. Advances in Neural Information Processing Systems, 32.

It would be nice to contrast the dual form in this paper with Stein discrepancies derived from the same function class (neural networks). There is also work that does this:

Ranganath, R., Tran, D., Altosaar, J., & Blei, D. (2016). Operator variational inference. Advances in Neural Information Processing Systems, 29.

Grathwohl, Will, et al. "Learning the stein discrepancy for training and evaluating energy-based models without sampling." International Conference on Machine Learning. PMLR, 2020.

* The connection I mentioned above seems to suggest that the proposed method can be recovered with minimizing the Stein discrepancy (with neural network function class) between variational posterior and true posterior. Could the authors comment on this?

* The literature review in section 1 is missing the work on implicit variational inference that uses gradient/score estimation techniques. See, e.g.,

Shi, J., Sun, S., & Zhu, J. (2018, July). A spectral approach to gradient estimation for implicit distributions. In International Conference on Machine Learning (pp. 4644-4653). PMLR.

Song, Yang, et al. "Sliced score matching: A scalable approach to density and score estimation." Uncertainty in Artificial Intelligence. PMLR, 2020.

* Appendix D suggests that UIVI is implemented with batch size 1, which means a substantially weaker baseline. I doubt if this is an unavoidable limitation of UIVI.

* Experiments do not include multi-modal target distributions.

**Summary Of The Paper:**

This paper presents a variational inference method with fisher divergence as the objective function to enable very flexible implicit variational posterior. The implicit variational posterior is defined as a hierarchical model that can be viewed as an infinite mixture (i.e., the semi-implicit variational posterior). The motivation for using fisher divergence is that one can get rid of computing the density of the hierarchical variational posterior. To achieve this, the paper proposes to rewrite the fisher divergence with its dual norm formulation. The final objective then has a minimax formulation, which results in an adversarial training algorithm.

**Summary Of The Review:**

Due to the weak points I raised, I currently lean towards rejection. However, I am happy to revise my opinion if these are sufficiently addressed.

---

> ### Author Response · Authors · 2022-11-19
> **Response to Reviewer cW7H**
>
> Thank you for your thoughtful review and valuable feedback. We address your specific questions and comments below
>
> Q1. The connection I mentioned above seems to suggest that the proposed method can be recovered with minimizing the Stein discrepancy (with neural network function class) between variational posterior and true posterior. Could the authors comment on this?
>
> A1. Thanks for mentioning the connection to stein discrepancies which is closely related to our formulation. As you have noticed above, our method can be viewed as minimizing the Stein discrepancy with neural network function class plus an L2 regularization term as follows
>
> $$
> \min_q\max_f \mathbb{E}_{x\sim q(x)}\left(\nabla \log p(x)^T f(x) + \nabla \cdot f  -  \frac12 \|f(x)\|^2\right)
> $$
>
> $$
> = \min_q\max_f \mathbb{E}_{x\sim q(x)} \left((\nabla \log p(x) - \nabla \log q(x))^T f(x) - \frac12 \|f(x)\|^2 \right)
> $$
>
> $$
> = \min_q \frac12 \mathbb{E}_{x\sim q(x)} \|\nabla \log p(x) - \nabla \log q(x)\|^2
> $$
>
> However, the stein discrepancy involves the $\nabla \cdot f$ term which is not easy to compute for high dimensional problems. Our method takes a further step by utilizing the hierarchical structure of $q(x)$ (the semi-implicit distributions). Using a mathematical trick that is similar to denoising score matching, we arrive at a formulation that easily scales up to high dimensions.
>
>
> Q2. The literature review in section 1 is missing the work on implicit variational inference that uses gradient/score estimation techniques.
>
> A2. Thanks for providing these related work. We will add them to the literature review in our revision.
>
> Q3. Appendix D suggests that UIVI is implemented with batch size 1, which means a substantially weaker baseline. I doubt if this is an unavoidable limitation of UIVI.
>
> A3: The key problem of UIVI is that it samples from the posterior distribution $q(z|x)$ to obtain the unbiased gradient estimate of the exact ELBO. Since HMC sampling of the posterior $q(z|x)$  requires computing the gradient $\nabla_{z}\log q(z|x)$ for the HMC update. Note that $\nabla_{z}\log q(z|x) = \nabla_{z}\log q(x|z) + \nabla_{z}\log q(z)$. Since $q(x|z)$ is the density obtained through neural network, its gradient with respect to $z$ has no explicit expression and can only be obtained through backpropagation. If UIVI uses a minibatch of $m$ data point $x_1, \cdots, x_m$, then it requires computing the Jacobian matrix
>
> $$
> [\nabla_{z}\log q(z|x_1)|_{z = z_1},
> $$
>
> $$
> \cdots,
> $$
>
> $$
> \nabla_{z}\log q(z|x_m)|_{z=z_m}]
> $$
>
> As far as we are concerned, the scalable calculation for this jacobian objective in Pytorch is not supported now.
>
> Q4. Experiments do not include multi-modal target distributions.
>
> A4. We have added an additional experiment on Bayesian neural networks in section 4.4, which have multi-modal posteriors.
>
> Q5. Section 3.1, "Although the hierarchical structure of q(x) allows us to estimate its score function via denoising score matching, the estimated score function would break the dependency ...". I don't understand this. Can you clarify?
>
> A5. Thanks for raising this up! In the original score matching objective, the score function $\nabla_x\log q_\varphi (x)$ depends on $\varphi$, so it will contribute to the gradient w.r.t. $\varphi$. However, if we use the estimated score function, it does not depend on $\varphi$ any more, the corresponding gradient computation therefore would be biased.

---

> > ### Author Response · Authors · 2022-12-04
> > **Additional notes to A1**
> >
> > Although the training objective are related, we want to emphasize a fundamental difference between our method and previous Stein discrepancy based variational inference methods: the approximation distribution $q$. Previously, Stein discrepance based VI methods are mainly particle-based methods **where $q$ is taken to be the empricial distribution of current particles**. As a result, these methods can only provide finite (which is the number of particles) samples for each run. **If new samples are required, one needs to retrain the particles which would be costly.** In this sense, particle-based VI methods is more like MCMC. On the other hand, semi-implicit VI is more like typical VI methods that **uses the semi-implicit variational distributions $q(x) = \int q(x|z) q(z) dz$ which allows instantaneously sampling once trained.**

---

> > ### Comment · Reviewer_cW7H · 2022-12-10
> > **Update after rebuttal**
> >
> > Hey,
> >
> > Apologize for being a bit late in updating my reviews. The rebuttal addressed my concerns, as detailed below:
> >
> > 1. I see that the key contributions here compared to the Stein operator formulation in Ranganath et al. (2016) is getting rid of the divergence of f term (which can be expensive to compute for large neural networks) using the semi-implicit formulation and a trick similar to that in denoising score matching. Having said that, it would be nice to demonstrate such improvements over Ranganath et al. (2016) in experiments (there is no need to compare to particle-based methods like SVGD).
> >
> > 2. The added BNN experiment introduces more complex posterior distributions than the unimodal ones in previous version and overall the method seems to perform better than SIVI.
> >
> > So I am increasing my score to acceptance.

---

### Decision · Program_Chairs · 2023-01-20

**Decision:**

Accept: notable-top-25%

**Justification For Why Not Higher Score:**

The reviewers have concerns on the extra computation and optimization difficulties brought by the min-max objective.

**Justification For Why Not Lower Score:**

This paper points out a new path to construct expressive variational posterior with tractable inference.

**Metareview: Summary, Strengths And Weaknesses:**

Following semi-implicit variational inference (SIVI), the paper proposes to use a semi-implicit variational distribution to approximate the posterior of model parameters. Different from SIVI that optimizes the semi-implicit variational distribution with an asymptotically-exact surrogate ELBO, the paper proposes to match the score of the posterior (i.e., gradient of the log posterior) with the score of the semi-implicit variational distribution. The key insight is that the semi-implicit construction allows the author to arrive at a tractable min-max objective for optimization. This derived objective, which cleverly combines SIVI and score matching, is new and novel. Experimental results on posterior inference for Bayesian (multinomial) logistic regression and Bayesian neural networks show that the proposed score-matching-based SIVI outperforms previous ELBO-based variational inference methods, including SIVI and UIVI.

**Note From Pc:**

if the above contains the word "oral" or "spotlight" please see: "oral" presentation means -> notable-top-5% and "spotlight" means -> notable-top-25%. As stated in our emails, we are disassociating presentation type from AC recommendations

**Summary Of Ac-Reviewer Meeting:**

The initial scores of the paper suggest this was a borderline paper that was worth a virtual meeting. The concerns raised in the meeting was that the proposed method has close relationship with Stein discrepancy and there is lack of comparison with Stein discrepancy-based variational inference algorithms. After discussion, the reviewers and AC came to the agreement that 1) while Stein discrepancy could lead to the score matching objective and could lead to the derivation of relevant variational inference algorithms, it requires the gradient of f that could be difficult to compute in high dimension, 2) The proposed min-max objective derived by utilizing the semi-implicit construction is simple and novel. Reviewers also pointed out other potentially better algorithms could be derived under Stein discrepancy, but the AC considers these alternatives as more appropriate for future work; in other words, whether these alternatives could work better/worse is not a necessary burden of this paper. The authors are encouraged to incorporate their rebuttal, in particular their responses on how their method is related to and different from Stein discrepancy, into their final version of the paper.